# Latent Adaptation of Foundation Policies for Sim-to-Real Transfer

**Longchao Da**[1], **Thirulogasankar Pranav Kutralingam**[1], **Lirong Xiang**[2] **& Hua Wei**[1] *

[1]Arizona State University, [2]Cornell University

`{longchao,tknolast,hua.wei}@asu.edu, lxiang@cornell.edu`

## Abstract

The sim-to-real problem remains a critical challenge in the real-world application of reinforcement learning (RL). The conventional sim-to-real methods heavily rely on resource-intensive re-training of the policy network to adapt to new domains, which limits the flexibility of the deployment of RL policies in ever-changing environments. Inspired by human locomotion, where individuals adjust their gait to new surface conditions without relearning the skill of walking, we introduce La-tent **Adapt**ation of **Found**ation Policies (**Found-adapt**), a framework that decouples this problem into skill acquisition and environment adaptation. Our method first pretrains a foundation policy on unlabeled offline trajectories from the source simulator, capturing diverse long-horizon behaviors as reusable skills. At deployment, instead of retraining the policy, we perform efficient latent space adaptation: a small amount of target-domain data is collected to refine a latent representation through an adapter network that incorporates parameter efficient alignment, which produces a task-ready controller under various system dynamics. This adaptation occurs entirely in the latent space, avoiding costly policy optimization while enabling robust transfer. Empirical results across multiple locomotion tasks and dynamic variations demonstrate that our method significantly reduces the sim-to-real gap. Further sensitivity analysis provides interesting insights into the requirements for data quality and applicable situations. These findings highlight how foundation policies with latent adaptation could serve as a general and efficient paradigm for real-world RL deployment. Implementation and experiment are available here.

## 1 Introduction

Deep Reinforcement Learning (DRL) has enhanced intelligent decision-making by trial-and-error fashion policy optimization. It can learn complex multi-tasks from video games (Shao et al., 2019) and sophisticated robotic manipulators (Nguyen & La, 2019) directly based on raw observations, which demonstrates superhuman performance in some environments with high-dimensional state and action spaces (Wang et al., 2016). However, real-world deployment of RL policies still lags behind the simulator successes (Da et al., 2024a); mainly because the policies trained in the simulator can rarely generalize across domains with different system dynamics, disturbances, and sensor noises, etc. (Da et al., 2025a), i.e., known as the Sim-to-Real Challenge (Wagenmaker et al., 2024).

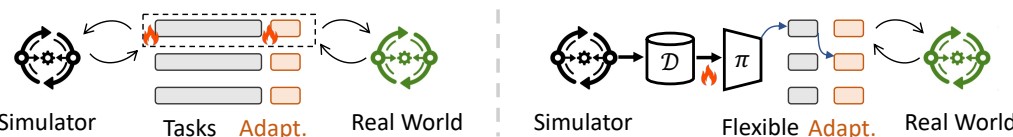

Figure 1: The difference between traditional sim-to-real adaptation (left) and the proposed method (right). Traditional sim-to-real involves two steps: task policy network training and task network adaptation, both of which are expensive and tightly bound. Proposed foundation policy $\pi$ enables flexible adaptation at an arbitrary task in any environment dynamics.

---

*Corresponding Author.

Bridging such sim-to-real gaps for RL policies remains a challenge in robotics, autonomous driving, and other safety-critical domains (James et al., 2019; Daza et al., 2023; Da et al., 2023a). Traditional approaches, such as domain randomization (Huber et al., 2024) and domain adaptation (Farhadi et al., 2024), typically rely on extensive online training for enough dynamics distribution coverage in the target domain: by either iteratively collecting real-world rollouts or deploying costly system identification procedures (Song et al., 2024), etc. This generally require substantial training resources to mitigate the gap. Thus, two fundamental challenges persist: (1) addressing new tasks requires retraining for task-specific policy prior to adaptation. (2) Adapting to new system dynamics necessitates re-training of the sim-to-real process, which incurs significant extra efforts and costs.

Inspired by (Sterelny, 2012), humans typically rely on previously acquired skills and adapt them to new environments without relearning from scratch. Instead of re-deriving motor primitives every time, people reuse a shared repertoire of skills and flexibly adjust them according to context and feedback. This analogy motivates the use of offline RL (Chen et al., 2024), where policies can be trained on diverse prior experience and later specialized to novel domains without costly online re-training. It serves as a foundation policy paradigm, which has offered a more generalizable policy acquisition by decoupling representation learning from task-specific policy training. By capturing generalist 'skillsets' from massive offline datasets, these methods can later instantiate near-optimal behaviors in a zero- or few-shot fashion to even unseen tasks (Luo et al., 2025), simply by conditioning on small amounts of prompts (Park et al., 2024b). Given the generalist potential, *for challenge (1)*, we proposed to leverage the foundation policies as the basic architecture for sim-to-real tasks to alleviate the re-training burden for policy networks. Then, *for challenge (2)*, to better tackle the sim-to-real gap, we propose **Found-adapt**, a latent space adaptation method that efficiently provides online adaptation with three steps: Cross-domain initial alignment, Dynamics signature extraction, and eventual Dynamics signature guided adaptation.

With only a small amount of target-domain data, our lightweight adapter estimates an appropriate latent representation that bridges the source and target dynamics. This enables effective sim-to-real adaptation without heavy retraining, while maintaining the ability to generalize across novel tasks. We have verified the effectiveness of the proposed method, which reveals great potential in real-world usage of RL policies. In conclusion, our contributions are threefold:

- Inspired by evolved apprentice, we propose to solve the sim-to-real challenge by leveraging the foundation policies with generalizability and get rid of task-specific retraining.

- We propose a novel and efficient adaptation method that leverages foundation policy representations and higher-order (meta) dynamics to derive the policy that bridges the source and target distributions during execution, improving the policies' sim-to-real ability.

- Broad Evaluation and Real-World Applicability. We demonstrate **Found-adapt** on multiple zero-shot RL tasks and environments, showcasing its strong transfer performance and practical potential by easily adapting to both new tasks and new environments.

## 2 RELATED WORK

**Sim-to-Real Methods in RL** There have been explorations to tackle the sim-to-real problems by interventions at different stages of the MDP process. In observation, there are practices like domain randomizations (Tobin et al., 2017; Tiboni et al., 2023), domain adaptation (Hu et al., 2022; Ho et al., 2021), and sensor fusions (Mahajan et al., 2024; Bohez et al., 2017), which intend to cover broader distribution during the policy training to overcome the potential shift. In action, there are works tackling the action latency (Dulac-Arnold et al., 2019; Dezfouli & Balleine, 2012) and quantifying the action uncertainties (Ilhan et al., 2021; Da Silva et al., 2020). Apart from the above, recent work intends to bridge the transition dynamics (Da et al., 2025a), a popular branch of work leverages grounded action to better calibrate the dynamics shift (Hanna & Stone, 2017; Da et al., 2024b), but all of these methods require extra training to shrink the domain gaps of the learned policies, which require extensive training on a mass of data. *Different* from these traditional methods, we leverage the pre-trained foundation models with existing and reusable skillsets, and design a way to efficiently prompt them to solve tasks in a specific domain with efficient computing overhead.

**Unsupervised Policy Pre-training** Previous studies have introduced a variety of unsupervised, task-agnostic objectives for pre-training a diverse set of policies, thereby speeding up subsequent

task learning. These methods pre-train policies with either exploration (Pathak et al., 2017; 2019; Mendonca et al., 2021; Rajeswar et al., 2023) or skill discovery objectives (Gregor et al., 2016; Eysenbach et al., 2018; Sharma et al., 2019; Klissarov & Machado, 2023; Park et al., 2023). *Different* from this direction, we focus on the offline setting, where we could learn and evaluate diverse policies purely from an offline dataset with unlabeled trajectories (Da et al., 2024c).

**Adaptive Policy Networks** Another line of work focuses on motor adaptation through end-to-end policy networks, such as Rapid Motor Adaptation (RMA) (Kumar et al., 2021) and Universal Policy with Online System Identification (UP-OSI) (Yu et al., 2017). A line of follow-up research inspired by RMA in real-world robotic systems includes bipedal locomotion (Kumar et al., 2022), in-hand object manipulation (Qi et al., 2023), manipulator control (Liang et al., 2024), and humanoid real-world policy adaptation (Hu et al., 2025), they perform adaptation by inferring latent dynamics from recent observations, specifically, they adapt policies to varying dynamics by jointly learning task objectives and online adaptation mechanisms, but they remain task-specific and require extensive online training, which limits their generalizability. *Different* from their method, we propose a generalizable sim-to-real foundation policy that can adapt to various tasks and system dynamics more flexibly and without per-task retraining.

## 3 PRELIMINARIES

**The Sim-to-Real Problem in RL** Following the classic definition (Sutton et al., 1998; Ding et al., 2020; Da et al., 2024b; Turnau et al., 2025), we formulate reinforcement learning in the standard Markov decision process setting (MDP): $M = (S, A, P, r, \mu, \gamma)$, where $S$ is the state space, $A$ the action space, $P : S \times A \to \Delta(S)$ refers to the transition probability. The reward function can be represented as $r : S \times A \times S \to \mathbb{R}$, $\mu$ the initial state distribution, and the discount factor is $\gamma \in (0, 1]$. A policy in RL, such as $\pi(a|s)$, defines a distribution over actions given state $s$, and its learning objective is to maximize the expected cumulative discounted reward $\mathbb{E}_\pi \left[ \sum_{t=0}^\infty \gamma^t r(s_t, a_t, s_{t+1}) \right]$. In practice, due to the cost of real-world exploration consequences, RL policies are usually trained in a simulator $E_{\text{sim}}$ and then executed in a real environment $E_{\text{real}}$ for testing or deployment (Salvato et al., 2021; Zhao et al., 2020). But since $E_{\text{sim}}$ always holds differences in dynamics compared to $E_{\text{real}}$, i.e., $P_{\text{sim}} \neq P_{\text{real}}$, thus, the policy often suffers a performance drop in $E_{\text{real}}$, which is denoted as the *Sim-to-Real gap*.

**Foundation Policies.** Inspired by foundation models in vision and language (Da et al., 2025b), a *foundation policy* is a generalist control policy trained from diverse offline trajectories, intended to capture a broad repertoire of reusable skills without specializing in a single downstream task. Given offline trajectories $\mathcal{D}$ collected from a source environment [1] $E_{\text{sim}}$, we define a *foundation policy* as a pair $(\phi, \pi)$, where $\phi : S \to \mathcal{Z}$ is a state encoder mapping states into a latent space $\mathcal{Z}$ and $\pi(a|s, z)$ is a latent-conditioned policy with $z \in \mathcal{Z}$. The encoder $\phi$ extracts reusable behavioral primitives, while varying $z$ spans a family of skills that can be composed to address different tasks. In this sense, foundation policies naturally support *task prompting*, where new rewards or goals are realized by selecting an appropriate latent. Moreover, their latent-conditioned structure also offers the potential for *system adaptation*, since adjusting $z$ provides a mechanism to align policy behavior with changes in environment dynamics. Our work develops this perspective by introducing latent adaptation techniques that unify task and system adaptation within the same framework.

## 4 METHODOLOGY

In this section, inspired by existing work (Park et al., 2024b), we will first introduce the learning of the policy model in Hilbert Space, as an instantiation for the foundation policies, and then we introduce details on how to adapt the latent space to solve the sim-to-real challenge while adapting towards various tasks based on the foundation policy's structure.

---

[1] In this paper, source env. = $E_{\text{sim}}$ and target env. = $E_{\text{real}}$, might be used interchangeably.

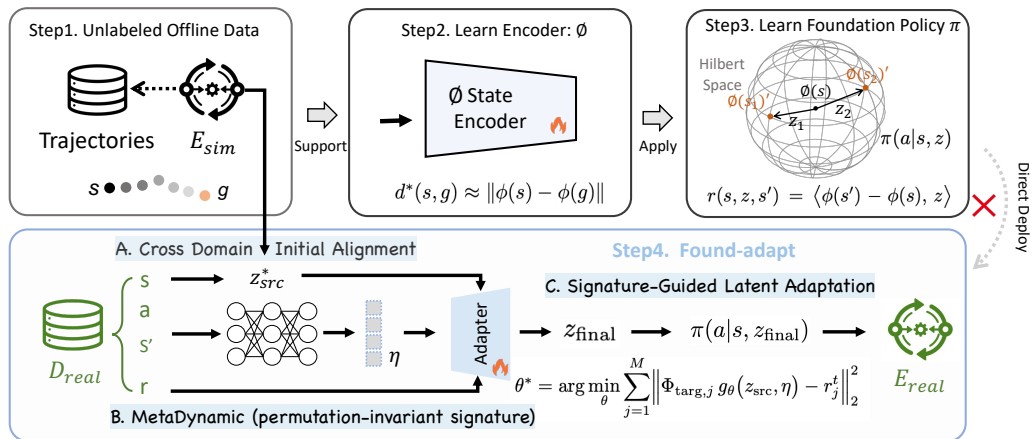

Figure 2: Overview of the proposed method. Offline trajectories from the simulator $E_{\text{sim}}$ train a state encoder $\phi$ and a latent-conditioned policy $\pi(a|s, z)$ using intrinsic rewards. Direct deployment degrades under dynamic gaps. We therefore perform *latent adaptation* with a small batch of target-domain data $D_{\text{tar}}$: (i) a weighted joint least-squares fit yields an initial latent $z_{\text{src}}^*$; (ii) a MetaDynamic network extracts permutation-invariant distributional features $\eta$; (iii) an adapter network refines $z_{\text{src}}^*$ into $z_{\text{final}}$. The refined latent conditions $\pi$ for robust execution in the target environment $E_{\text{tar}}$ without retraining the policy.

## 4.1 FOUNDATION MODEL IN HILBERT SPACE

Based on the definition in Section 3, one concrete instantiation of a foundation policy is given by embedding states into a Hilbert space. The key idea, introduced by (Park et al., 2024b), is that distances in this space can be aligned with temporal relations in trajectories, which makes the representation suitable for skill composition and task adaptation. Formally, an encoder $\phi : S \to \mathcal{Z}$ maps states into a latent space $\mathcal{Z} = \mathbb{R}^D$ with Euclidean inner product. The encoder is trained so that $\|\phi(s) - \phi(g)\|$ approximates the true temporal distance between $s$ (certain state) and $g$ (goal), thus capturing the long-horizon structure of trajectories.

Given such an encoder, a latent-conditioned policy $\pi(a|s, z)$ with $z \in \mathcal{Z}$ is trained using an intrinsic reward $r(s, z, s') = \langle \phi(s') - \phi(s), z \rangle$. The intrinsic reward aligns policy transitions with the latent vector $z$, ensuring that for any direction $z \in \mathcal{Z}$ the policy can induce state changes consistent with $z$ (thereby forming a set of directional primitives), and the collection $\{\pi(\cdot|s, z) : z \in \mathcal{Z}\}$ can be viewed as a skill family that spans the embedding. For downstream tasks, one can solve for an appropriate $z^*$ by aligning the latent prediction with the task reward, which reduces to a least-squares regression on offline samples. In the special case of goal reaching, $z^*$ simplifies to the normalized vector between state $\phi(s)$ and goal $\phi(g)$, yielding a closed-form solution.

This enables task-level adaptation without retraining the policy parameters. However, the solution implicitly assumes that the transition statistics used for representation learning in $E_{\text{sim}}$ remain consistent at deployment in $E_{\text{real}}$. In practice, domain shift in dynamics often invalidates this assumption, leading to degraded performance. To address this sim-to-real gap, we next introduce a latent adaptation mechanism that augments the above regression with additional components for robust cross-dynamics deployment.

## 4.2 LATENT ADAPTATION FOR SIM-TO-REAL TRANSFER

**Baseline latent vector $z^*$ for foundation policy.** To deploy a Hilbert foundation policy on a new task, one solves for the optimal latent vector as follows:

$$z^* = \arg \min_{z \in \mathcal{Z}, \|z\|=1} \mathbb{E}_{(s,a,s') \sim \mathcal{D}} \left[ \left( R(s, a, s') - \langle \phi(s') - \phi(s), z \rangle \right)^2 \right] \quad (1)$$

where $R(s, a, s')$ is the task reward and $\phi(s)$ is the Hilbert embedding from Section 4.1. In practice, given samples $\{(s_i, a_i, s'_i, r_i)\}_{i=1}^N$, let $\Phi = [\phi(s'_1) - \phi(s_1), \ldots, \phi(s'_N) - \phi(s_N)]^\top \in \mathbb{R}^{N \times D}$ and

$r = [r_1, \ldots, r_N]^\top \in \mathbb{R}^N$. Eq. 1 reduces to the standard least-squares problem:

$$\hat{z} = \arg \min_{z \in \mathbb{R}^D} \|\Phi z - r\|_2^2 \qquad (2)$$

which yields the closed form solution: $\hat{z} = (\Phi^\top \Phi)^{-1} \Phi^\top r$, derivation is shown in Appendix C. After normalization to unit norm, we obtain $z^* = \hat{z}/\|\hat{z}\|$. For goal reaching tasks with target $g$, this further simplifies to the normalized vector $z^* = (\phi(g) - \phi(s))/\|\phi(g) - \phi(s)\|$.

**The performance gap introduced by the dynamics gap.** The baseline inference in Eq. 1 assumes that the transition statistics used during training remain valid at deployment. Formally, let the simulator dynamics be $P_{\text{sim}}(s'|s, a)$ and the real dynamics be $P_{\text{real}}(s'|s, a)$. We denote their discrepancy as $\Delta_P(s'|s, a) = P_{\text{real}}(s'|s, a) - P_{\text{sim}}(s'|s, a)$. This difference propagates into the feature matrices, since each row is defined by the embedding difference $\phi(s') - \phi(s)$. Thus, if $\Phi_{\text{sim}}$ is the feature matrix computed under $P_{\text{sim}}$, then under real dynamics we have $\Phi_{\text{real}} = \Phi_{\text{sim}} + \Delta_\Phi{}^2$, where $\Delta_\Phi$ collects the deviations induced by $\Delta_P$.

The least-squares estimate under simulator data is $\hat{z}_{\text{sim}} = (\Phi_{\text{sim}}^\top \Phi_{\text{sim}})^{-1} \Phi_{\text{sim}}^\top r$, whereas the true optimal solution under real dynamics is $\hat{z}_{\text{real}} = (\Phi_{\text{real}}^\top \Phi_{\text{real}})^{-1} \Phi_{\text{real}}^\top r$. Substituting $\Phi_{\text{real}} = \Phi_{\text{sim}} + \Delta_\Phi$ reveals that $\hat{z}_{\text{real}} \neq \hat{z}_{\text{sim}}$, and the difference $\Delta_z = \hat{z}_{\text{real}} - \hat{z}_{\text{sim}}$ shows the *dynamics gap* at the level of the latent solution. Without explicitly correcting for $\Delta_\Phi$, deploying $\pi(a|s, \hat{z}_{\text{sim}})$ in $E_{\text{real}}$ will generally lead to degraded performance.

**A. Cross-Domain Initial Alignment.** To mitigate this gap, One possible way to address this issue is to re-train the state encoder $\phi$ using additional trajectories from the target environment $E_{\text{real}}$, i.e., constructing a new dataset $\mathcal{D}_{\text{tar}} = \{(s_j^t, a_j^t, s_j'^t, r_j^t)\}$ and updating $\phi$ so that it aligns with $P_{\text{real}}$. However, such re-training is computationally intensive and reduces the modularity of foundation policies by entangling representation learning with deployment. Instead, we propose to keep $\phi$ fixed as learned from $E_{\text{sim}}$ and directly adapt the latent variable $z$ toward $E_{\text{real}}$ by leveraging $\mathcal{D}_{\text{tar}}$. Concretely, this requires only solving regression problems of the form $\min_z \|\Phi_{\text{sim}} z - r_{\text{sim}}\|^2 + \lambda \|\Phi_{\text{tar}} z - r_{\text{tar}}\|^2$ and performing lightweight parameter updates for an adapter at inference time. This procedure avoids costly re-optimization of $\phi$, yet yields $z_{\text{final}}$ that conditions $\pi(a|s, z_{\text{final}})$ for robust deployment in $E_{\text{real}}$.

Since we tend to consider samples from both the simulator and the real conditions, we extend the least-squares step. Suppose we have $D_{\text{src}} = \{(s_i, a_i, s_i', r_i)\}_{i=1}^N$ from the simulator and $D_{\text{tar}} = \{(\tilde{s}_j, \tilde{a}_j, \tilde{s}_j', \tilde{r}_j)\}_{j=1}^M$ from the real environment. We form the feature matrices $\Phi_{\text{sim}} = [\phi(s_i') - \phi(s_i)]_{i=1}^N \in \mathbb{R}^{N \times D}$ and $\Phi_{\text{tar}} = [\phi(\tilde{s}_j') - \phi(\tilde{s}_j)]_{j=1}^M \in \mathbb{R}^{M \times D}$, with corresponding reward vectors $r_{\text{sim}} \in \mathbb{R}^N$ and $r_{\text{tar}} \in \mathbb{R}^M$. We then solve the weighted regression as follows:

$$z_{\text{src}} = \arg \min_{z \in \mathbb{R}^D} \|\Phi_{\text{sim}} z - r_{\text{sim}}\|_2^2 + \lambda \|\Phi_{\text{tar}} z - r_{\text{tar}}\|_2^2 \qquad (3)$$

where $\lambda > 1$ biases the fit toward real data. This admits the closed form $z_{\text{src}} = (A^\top A)^{-1} A^\top b$ with $A = \begin{bmatrix} \Phi_{\text{sim}} \\ \sqrt{\lambda} \Phi_{\text{tar}} \end{bmatrix}$ and $b = \begin{bmatrix} r_{\text{sim}} \\ \sqrt{\lambda} r_{\text{tar}} \end{bmatrix}$, followed by normalization. This weighted joint regression integrates information from both domains, allowing the latent solution to provide an initial, partially corrected dynamics gap while preserving the efficiency of closed-form inference.

**B. MetaDynamic - A Permutation-Invariant Network for Dynamics Signature.** While cross-domain alignment addresses first-order differences in transition features, it may fail to capture higher-order variations such as distributional shifts in state visitation or structural patterns in dynamics. To address this, we compute a permutation-invariant summary of the target-domain:

$$\eta = \text{MetaDynamic}(\{\phi(\tilde{s}_j)\}_{j=1}^M) \in \mathbb{R}^K \qquad (4)$$

where $\text{MetaDynamic}$ is a set-encoding function trained on simulator data and frozen at deployment. The permutation invariance serves as a critical component, since $\{\phi(\tilde{s}_j)\}_{j=1}^M$ is an unordered collection of states sampled from trajectories rather than a sequence with fixed ordering; the representation

---

[2]Here $\Phi$ denotes a feature matrix derived from the encoder $\phi$; it is not an additional learnable parameter.

must depend only on the empirical distribution of states, not on sample order. We interpret $\eta$ as a *dynamics signature* of the target domain, i.e., a compact descriptor encoding statistical regularities of $E_{\text{real}}$ beyond mean feature alignment. This signature serves as an auxiliary input to refine latent adaptation, enabling the policy to exploit structural information about the target dynamics without retraining the encoder $\phi$. The details of the $\mathrm{MetaDynamic}$ structure and training configurations are provided in the Appendix. G.

**C. Signature-Guided Latent Adaptation.** Although cross-domain alignment partially reduces the dynamics gap, the latent solution $z_{\text{src}}$ may still drift when directly applied in $E_{\text{tar}}$. To further adapt, we introduce a parametric adapter $g_\theta : \mathbb{R}^D \times \mathbb{R}^K \to \mathbb{R}^D$ that leverages both the initial solution $z_{\text{src}}$ and the dynamics signature $\eta$. The adapter is initialized to approximate the identity mapping, i.e., $g_\theta(z_{\text{src}}, \eta) \approx z_{\text{src}}$, ensuring stability at the start of adaptation. We then update $\theta$ using the target-domain dataset $\mathcal{D}_{\text{tar}}$, updating the parameters for a small number of gradient steps by minimizing below, the loss $\ell_j$ conditions on the solution $z_{\text{src}}$ and the dynamics signature $\eta$:

$$\mathcal{L}(\theta) = \frac{1}{M} \sum_{j=1}^{M} \underbrace{\left\| \Phi_{\text{tar},j}\, g_\theta(z_{\text{src}}, \eta) - \tilde{r}_j \right\|_2^2}_{\ell_j(\theta; z_{\text{src}}, \eta)}. \tag{5}$$

where $\Phi_{\text{tar},j}$ is the $j$-th row of the target-domain feature matrix and $\tilde{r}_j$ is the corresponding reward. This produces an adapted parameter set $\theta^*$ and yields a refined latent vector:

$$z_{\text{final}} = \sqrt{D}\, \frac{g_{\theta^*}(z_{\text{src}}, \eta)}{\|g_{\theta^*}(z_{\text{src}}, \eta)\|} \tag{6}$$

The resulting $z_{\text{final}}$ incorporates both cross-domain alignment and distributional structure, and conditions the foundation policy $\pi(a|s, z_{\text{final}})$ for robust execution in $E_{\text{tar}}$ without retraining the policy.

In summary, the proposed latent adaptation provides a parameter-efficient approach to sim-to-real transfer. From the perspective of conventional RL policies, which typically require costly policy $\pi$ retraining on target-domain data, our method is lightweight and avoids modifying the policy network altogether. From the perspective of foundation policies, which were originally conceived for task prompting under fixed dynamics, our method repositions them as a versatile paradigm that can adapt not only across tasks but also across environments with differing dynamics, thus transforming their role from intra-domain generalization to a unified framework for sim-to-real transfer.

### 4.3 Overall Framework

As shown in Fig. 2, our framework reframes foundation policies to address the challenging sim-to-real transfer problem through prompting and parameter-efficient domain adaptation. We begin with offline trajectories collected from the simulator $E_{\text{sim}}$, which are used to train a state encoder $\phi$ and a latent-conditioned policy $\pi(a|s, z)$. This training stage produces a foundation policy that encapsulates a repertoire of reusable skills indexed by latent variables $z$, supporting flexible task prompting without modifying the policy parameters. However, direct deployment in a target environment $E_{\text{tar}}$ is hindered by dynamics gaps that render simulator-derived latents suboptimal. To overcome this, our latent adaptation procedure adjusts $z$ while keeping $\pi$ fixed: we propose (i) cross-domain regression yields an initial $z_{\text{src}}$, (ii) a permutation-invariant MetaDynamic encodes a dynamics signature $\eta$, and (iii) a signature-guided adapter refines $z_{\text{src}}$ into $z_{\text{final}}$. The resulting $z_{\text{final}}$ conditions the frozen foundation policy for robust execution in $E_{\text{tar}}$. This parameter-efficient design redefines the scope of foundation policies, elevating them from intra-domain task generalization to a unified paradigm for sim-to-real deployment.

## 5 Experiments

**Dataset and Tasks:** In this section, we experiment on environments from a widely adopted benchmark in offline RL (Park et al., 2024a). We test different methods' sim-to-real transferability using the sim-to-sim setting, which is a common practice to verify the methodology for better reproducibility (Da et al., 2025a; Zhao et al., 2020; James et al., 2019), we treat the default environment setting

| Setting | Methods | Task's Average Return (Sim-to-Real Gap) | | Average Time Cost | Hilbert Space Embeddings |
|---|---|---|---|---|---|
| | | Stand($\Delta \uparrow$) | Walk($\Delta \uparrow$) | ($\downarrow$) | $\phi(s)$ under dynamics perturbs |
| $E_{sim}$ | Foundation Policy | $887.61_{\pm 18.93}$ | $764.99_{\pm 25.73}$ | $0.73_{\pm 0.03}$ |  |
| $G1$ | Direct-Transfer | $494.24\,(-393.37)_{\pm 95.89}$ | $318.93\,(-446.06)_{\pm 127.15}$ | $\mathbf{5.06\,s \pm 0.05}$ | |
| | Vanilla-GAT | $376.27\,(-511.34)_{\pm 101.65}$ | $185.66\,(-579.33)_{\pm 80.22}$ | $7.59\,s \pm 1.70$ | |
| | UGAT | $371.44\,(-516.17)_{\pm 187.32}$ | $326.49\,(-438.50)_{\pm 48.09}$ | $9.15\,s \pm 2.50$ | |
| | PAD | $\underline{557.03}\,(-330.58)_{\pm 33.79}$ | $\underline{448.50}\,(-316.49)_{\pm 51.61}$ | - | |
| | **Found-adapt** | $\mathbf{562.72}\,(-202.27)_{\pm 41.17}$ | $\mathbf{472.25}\,(-292.74)_{\pm 42.44}$ | $\underline{6.22\,s \pm 0.12}$ | |
| $G2$ | Direct-Transfer | $222.49\,(-665.12)_{\pm 27.40}$ | $150.98\,(-614.01)_{\pm 29.73}$ | $\mathbf{5.36\,s \pm 0.11}$ | |
| | Vanilla-GAT | $60.57\,(-827.04)_{\pm 14.87}$ | $71.96\,(-693.03)_{\pm 54.30}$ | $7.56\,s \pm 2.30$ | |
| | UGAT | $175.42\,(-712.19)_{\pm 91.35}$ | $163.25\,(-601.74)_{\pm 90.00}$ | $9.10\,s \pm 2.90$ | |
| | PAD | $\mathbf{273.16}\,(-614.45)_{\pm 54.60}$ | $\mathbf{189.17}\,(-575.82)_{\pm 2.53}$ | - | |
| | **Found-adapt** | $\underline{231.75}\,(-655.86)_{\pm 34.59}$ | $\underline{182.46}\,(-582.53)_{\pm 20.89}$ | $\underline{6.11\,s \pm 0.07}$ | |
| $G3$ | Direct-Transfer | $213.15\,(-674.46)_{\pm 78.96}$ | $105.00\,(-659.99)_{\pm 10.80}$ | $\mathbf{5.14\,s \pm 0.10}$ | |
| | Vanilla-GAT | $32.28\,(-855.33)_{\pm 12.00}$ | $28.80\,(-736.19)_{\pm 10.83}$ | $7.71\,s \pm 2.70$ | |
| | UGAT | $72.30\,(-815.31)_{\pm 42.51}$ | $16.45\,(-748.54)_{\pm 1.08}$ | $9.10\,s \pm 1.70$ | |
| | PAD | $\underline{262.21}\,(-625.40)_{\pm 33.33}$ | $\mathbf{127.96}\,(-637.03)_{\pm 17.83}$ | - | |
| | **Found-adapt** | $\mathbf{322.06}\,(-565.55)_{\pm 35.18}$ | $\underline{120.87}\,(-644.12)_{\pm 8.42}$ | $\underline{6.08\,s \pm 0.11}$ | |
| $G4$ | Direct-Transfer | $63.81\,(-823.80)_{\pm 14.14}$ | $33.03\,(-731.96)_{\pm 6.82}$ | $\mathbf{5.28\,s \pm 0.09}$ | |
| | Vanilla-GAT | $57.83\,(-829.78)_{\pm 20.27}$ | $25.42\,(-739.57)_{\pm 11.53}$ | $7.61\,s \pm 2.50$ | |
| | UGAT | $\mathbf{80.33}\,(-807.28)_{\pm 47.76}$ | $14.51\,(-750.48)_{\pm 0.45}$ | $8.63\,s \pm 2.40$ | |
| | PAD | $\underline{78.69}\,(-808.92)_{\pm 9.47}$ | $\underline{37.13}\,(-727.86)_{\pm 9.70}$ | - | |
| | **Found-adapt** | $71.70\,(-815.91)_{\pm 11.81}$ | $\mathbf{42.57}\,(-722.42)_{\pm 104.54}$ | $\underline{6.12\,s \pm 0.11}$ | |

Table 1: Performance under *Gravity* variations comparing baseline methods, and **Found-adapt**. Each cell shows the average return with the sim-to-real gap relative to $E_{sim}$ and the standard deviation over 5 runs. Higher is better for return, and lower is better for time cost. The best average per column is shown in **bold**, and the second best is underlined. PAD runtime is marked as '-', indicating not directly comparable (hour-scale). The right column shows 2D t-SNE of the encoder embeddings $\phi(s)$ under gravity perturbations ($G_0$, $G_1$, and $G_4$). As the dynamics perturbation increases, the manifold structure observed in $G_0$ progressively distorts, revealing representation drift under perturbed environments, more comprehensive results are shown in Appendix, Fig. 7.

as $E_{\text{sim}}$ and control various $E_{\text{real}}$ by designing high fidelity system dynamics following the configurations in Table 3. In brief, there are two sets of variables of settings: friction and gravity, which are applied at each joint and affect degree of freedom, to reflect high-fidelity system dynamics.

**Baselines:** We compared with several classic and advanced baselines to provide empirical insights. The first baseline is `Direct-Transfer`, which is the foundation policy learned in $E_{sim}$ and directly deployed to $E_{real}$. Then, we compared with the grounded action transformation (`GAT`) (Hanna & Stone, 2017), and uncertainty-based GAT (`UGAT`) (Da et al., 2023b), which is developed by post-training a grounding module for the foundation policy to ground the actions provided by the foundation policy, and UGAT integrates an evidential layer to derive the uncertainty and use it to reject low-quality actions. Besides, we provide four versions of **Found-adapt** for the ablation study: `F(init)` (closed form solution $\pi(a|s, z_{\text{src}}^{*})$ by cross-domain initial alignment from Sec. 4.2A), `F(dyna)` ($z_{\text{final}}^{*}$ directly calculated from MetaDynamic network in Sec. 4.2B), `F(init, dyna)` (make use of both initial alignment and meta dynamics signature, but without online adaptation.) and `F-all` (i.e., full model of **Found-adapt**). There are methods in deployment adaptation branch, however, this branch of methods are not adaptive to different tasks, thus, it needs to be pretrained on specific tasks then perform deployment adaptation individually, to represent this research direction, we implemented a classic baseline: Policy Adaptation during Deployment (PAD) by (Hansen et al.), given the high time cost, we only provide magnitude in evaluation.

**Evaluation Metrics:** The primary goal of this work is to mitigate the performance gap of the derived policy $\pi_\theta$ in the simulation environment $E_{sim}$ and in the real-world environment $E_{real}$, thus we calculate the performance difference $\Delta$ for the episode return of each task. We denote their differences as $\Delta_{\text{Return}}$. For a given metric '*Episode Return*' in two domains. Normally, $R_{real} < R_{sim}$, thus the gap $\Delta_{\text{Return}}$ is negative, and the higher, the smaller the gap is ($\uparrow$), applied in Table. 1.

$$\Delta_{\text{Return}} = R_{real} - R_{sim} \qquad (7)$$

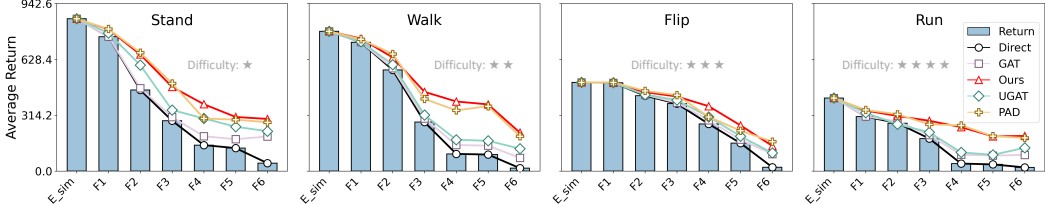

Figure 3: The performance on the **friction-based settings** (F1 - F6). In this image, from left to right, the blue bar shows the performance drop in the original system dynamics ($E_{sim}$) along with the increase of the difficulty of the system (reflected by the friction severeness as in Table. 3). The foundation policy can adapt to various tasks, and our method shows best sim-to-real gap mitigation.

## 5.1 EXPERIMENT ANALYSIS

**(1) The performance comparison among the baselines.** In this section, we tend to verify the performance of our method in comparison to baselines. In Table 1, we compared **Found-adapt** with four baselines under various gravity changes, including three types: Direct Transfer, grounded action transformation (GAT, UGAT), and deployment adaptation (PAD). It is worth noting that deployment adaptation methods require pretraining on task-specific data collections; thus, this branch of method can not provide an absolute fair comparison to the task: 'Adapt policy to various tasks and multiple system dynamics at mean time without further training'. More details please visit Appendix E (3). We evaluate on 4 different tasks in the Walker environment, comparing different methods' performance (return) and sim-to-real gap as in brackets (the higher the better), meanwhile, we also compare the time cost during online adaptation. Our method consistently ranked at the top 2 performing methods. Sometimes it is slightly worse than the PAD; however, PAD requires task-specific pretraining and updates are coupled to downstream behavior, whereas our method leverages a task-agnostic foundation representation shared across tasks: stand, flip, etc. We also perform a comprehensive evaluation on friction settings, as shown in Fig. 3, from stand to run. As the difficulty increases, the overall performance drops; however, as shown in the red line, our method successfully makes an improvement from the blue bar (which is the direct transfer performance in $E_{real}$). The adaptation process with the latent $z$ representation changes is shown in the Fig. 5 (b).

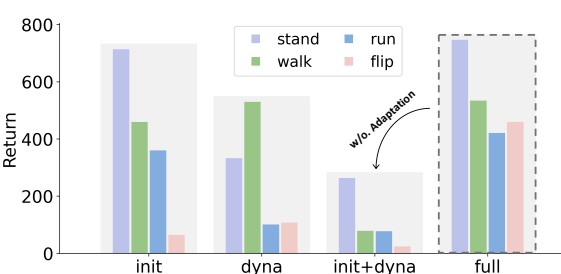

(a) The ablation study of the **Found-adapt** on the friction F1 setting. Dotted box shows full model improves across tasks, and adaptation is necessary, as in `init+dyna` (w/o. adapt).

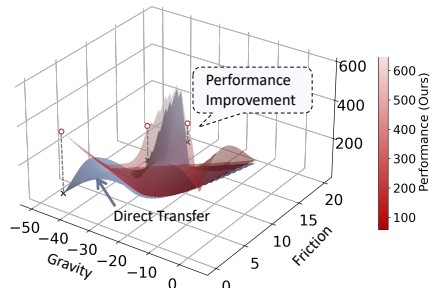

(b) The landscape on `Walk` of both factors to analyze the difficulty of the optimization task, our method provides consistent improvement.

Figure 4: The partial results of case studies on the ablation and sensitivity of proposed **Found-adapt**.

**(2) The ablation study of the proposed Found-adapt.** The proposed method mainly has three components: Initial alignment solution $z_{src}$, MetaDynamics network with signature $\eta$, and Latent adapter $g_\theta$. (Sec.4.2 A-C). We perform the ablation study to understand the contribution of different components in the model's result. As shown in Fig. 4 (a), **F(init)** uses only $z_{src}$, it can score well on the easy *stand* task but exhibits high variance on other tasks because it fails to capture higher-order dynamics features; **F(dyna)** uses only dynamic signature $\eta$, underperforms since the sole signature struggles to produce a valid latent $z$; interestingly, the **F(init+dyna)** performs worst, since it applies simple merge with $z_{src}$ and $\eta$, as the latent mixes uncalibrated cues from alignment and higher-order dynamics, it fails to exploit either; once we add the adapter $g_\theta$ (Eq. 5), **F(all)** - ours: yields consistent

gains on all tasks (dotted box) by reconciling $z_{\text{src}}$ with $\eta$, reducing bias and variance, and turning the dynamics signature into extra corrections, more results are in Fig. 8.

**(3) Study of the system variable sensitiveness and the model's ability.** We analyze how performance changes with friction and gravity jointly on `Walk` task. The results in Fig. 4 (b) show a two-factor response surface where the adapted policy (gradient red surface) consistently lies *above* direct transfer (blue), indicating non-negative lift across the domain. The gap (improvement) is most visible in harder ranges: strong gravity shifts (-34 to -44) combined with moderate friction,

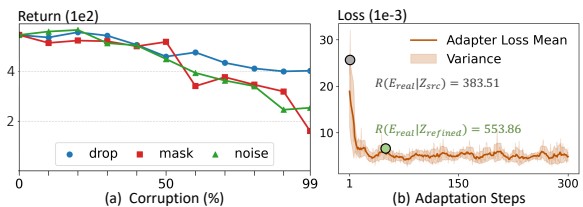

Figure 5: The (1) data quality analysis under three modes, and (2) latent vector evolution $z_{\text{src}} \rightarrow z_{\text{refined}}$.

suggesting that adaptation receives larger gains where the sim-to-real gap is wide. Moving toward the nominal dynamics (gravity $\approx -9.81$, friction $\approx 1$), the two surfaces converge, but ours remains slightly higher, evidencing reliable improvement even in mild conditions. Overall, despite the non-linear and rugged landscape, the adapted model maintains robustness and yields the largest benefits in adverse regimes.

**(4) The analysis of the requirement of target-domain data.** In collected target-domain data, $\mathcal{D}_{\text{tar}} = \{(s_j, r_j, s'_j)\}_{j=1}^{M}$. We corrupt *transitions* with a rate $p\%$ that selects an index set $J \subset \{1, \ldots, M\}$. We study three modes that mimic common data issues: (i) `drop`: remove the selected tuples $\{(s_j, r_j, s'_j)\}_{j \in J}$ from $\mathcal{D}_{\text{tar}}$ (*data sparsity*); (ii) `mask`: keep length but set $s_j = \mathbf{0}$, $s'_j = \mathbf{0}$, and $r_j = 0$ for $j \in J$ (*bad imputation / 'missing-as-zero' bias*); (iii) `noise`: add Gaussian perturbations scaled by the empirical per-dimension standard deviation to the raw signals, i.e., $s_j \leftarrow s_j + \epsilon_1, s'_j \leftarrow s'_j + \epsilon_2, r_j \leftarrow r_j + \epsilon_3$ with $\epsilon_1, \epsilon_2, \epsilon_3 \sim \mathcal{N}(0, \sigma^2)$, simulate the *label noise* scenarios. We evaluate on the consistent `Walker` environment, *stand* task under gravity G1 in Table. 3. As shown in the Fig. 5 (a), ***All three modes only suffer slightly with drops when the corruption is less than 50%***, showing our method is relatively robust to data quality. Then, Performance degrades most *gracefully* under `drop`, indicating our method could capture the dynamics signature from little data with high efficiency. `mask` fails early: means zero-filled values inject systematic bias, and hinders the adaptation, pulling the latent toward degenerate solutions once $\gtrsim 50 - 60\%$ of data are masked. The `noise` hurts consistently under heavy corruption ($\gtrsim 50\%$). In practice, when data are suspect, it is suggested to discard rather than zero-filling when applying our method, keep noise low, and consider denoise objectives if noisy data are unavoidable.

**(5) The relationship of the adaptation loss and the sim-to-real ability.** This study aims to understand how the adaptation process advances the sim-to-real ability of our method. We treat the 'Stand' task as an example. In the process, we leveraged the pre-trained foundation policy, and applied our method; then, we recorded the loss changes (step-wise) during the adaptation of network $g_\theta$ as in Eq. 5, by this, we derive the $\pi_i$, $i \in [1, 60]$ from 60 steps. We test each $\pi_i$'s performance and calculate the improvement as in Eq. 7. To better illustrate the correlation, we take a negative value of the loss along the x-axis and then normalize both dimensions. On left side of Fig. 6, the grey area shows the 95% confidence interval around the fitted regression line, on the right side, we show the KDE of the joint distribution to move beyond linear analysis, it

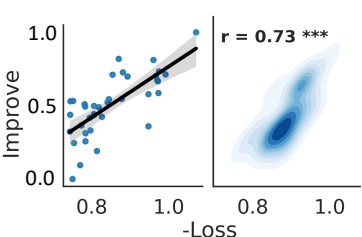

Figure 6: Correlation of the adaptation loss and the target domain $E_{real}$ performance improvement.

shows a high-density ridge aligned with the regression (darker $\rightarrow$ denser), confirming monotonicity and the scarcity of cases where lower loss yields better performance ('****' indicates p $\leq 0.001$). This study reveals that: as '$-$loss' increases (i.e., the better adaptation results), the sim-to-real performance consistently grows.

# 6 CONCLUSION

This work presents the study of leveraging foundation policies for adaptive sim-to-real transfer. The framework pre-trains a versatile foundation policy from offline simulation data and then employs a lightweight latent adapter, inferred from limited target-domain samples, to align with real-world dynamics efficiently. This design enables rapid adaptation without costly retraining, mirroring how humans reuse previously acquired skills and flexibly adjust them to novel environments.

Empirical evaluations across diverse locomotion tasks and varying dynamics factors, such as gravity and friction, demonstrate that our method mitigates domain gaps and achieves robust transfer compared to conventional approaches. Meanwhile, we acknowledge two limitations: (i) the degree of mitigation varies across environments, and (ii) sensitivity to hyperparameters may require careful tuning for different tasks; the prompt quality also affects generalization performance in the target domain. These show opportunities for refinement through adaptive weighting, uncertainty-aware objectives, and broader real-world extensions.

## ACKNOWLEDGEMENT

The work was partially supported by NSF award #2442477. We thank Amazon Research Awards, Cisco Research Awards, Google, and OpenAI for providing us with API credits. The authors acknowledge Research Computing at Arizona State University for providing computing resources. The views and conclusions in this paper should not be interpreted as representing any funding agencies.

## ETHICS STATEMENT

This research focuses on the sim-to-real adaptation challenge in reinforcement learning, which is fundamental and critical to the real-world application of RL. By solving the sim-to-real challenge, this work will be useful for mitigating potential real-world policies' risks, such as autonomous driving, etc. In this research itself, there is no ethical concern.

## REPRODUCIBILITY STATEMENT

The implementation details and instructions needed to reproduce the main experimental results are included in the supplementary material. Implementation and experiment are available at here.

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

APPENDIX

## A    LLM USAGE STATEMENT

In accordance with the ICLR 2026 submission guidelines, we disclose that large language models (LLMs) were used only for language editing and grammar checking. No LLMs were employed for generating research ideas, designing methodologies, producing experimental results, or creating data. All scientific content, analysis, and conclusions were developed and verified by the authors.

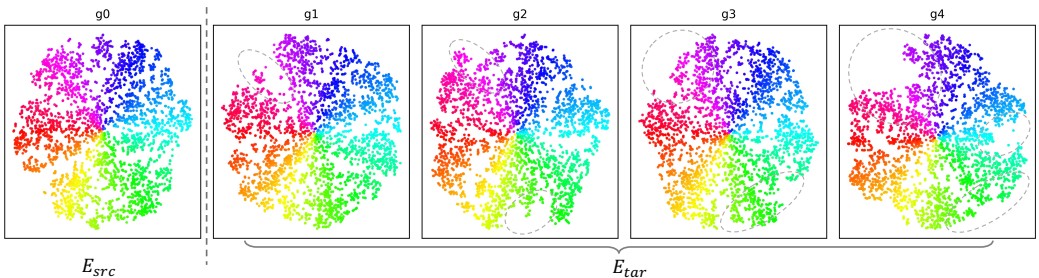

Figure 7: The 2D t-SNE visualization of Hilbert-space embeddings for `Walk` environment across default and 4 gravity settings: $\phi(s)$ under increasing dynamics perturbations. From $g_0$ (default standard gravity) to $g_4$ (strongly perturbed dynamics), the embedding structure gradually collapses: the clean manifold geometry in $g_0$ becomes increasingly distorted and contracted as dynamics shift grows. This illustrates that the state-visitation distribution under perturbed dynamics diverges significantly from the simulator distribution, producing noticeable drift in the fixed encoder's feature space, which our adaptation module is designed to correct the policy on that environment.

## B    HILBERT SPACE PRELIMINARIES

**Definition B.1** (Real Hilbert Space). A *real Hilbert space* is a real vector space $\mathcal{Z}$ equipped with an inner product as $\langle \cdot, \cdot \rangle : \mathcal{Z} \times \mathcal{Z} \to \mathbb{R}$, such that the induced norm $\|z\| = \sqrt{\langle z, z \rangle}$ makes $\mathcal{Z}$ into a complete metric space under the distance $d(z, z') = \|z - z'\|$. Completeness means that every Cauchy sequence in $\mathcal{Z}$ converges (with respect to $d$) to a point in $\mathcal{Z}$.

In our approach, we rely on the geometry of a real Hilbert space $(\mathcal{Z}, \langle \cdot, \cdot \rangle)$. Below, we briefly introduce the key definitions and properties:

### B.1    INNER PRODUCT

For any $z, w \in \mathcal{Z}$, the inner product is $\langle z, w \rangle = \sum_{i=1}^{D} z_i w_i$, which satisfies:

- **Symmetry:** $\langle z, w \rangle = \langle w, z \rangle$.
- **Linearity:** $\langle \alpha z + z', w \rangle = \alpha \langle z, w \rangle + \langle z', w \rangle$.
- **Positive definiteness:** $\langle z, z \rangle \geq 0$ with equality iff $z = 0$.

### B.2    NORM

The norm induced by the inner product measures the "length" of a vector: $\|z\| = \sqrt{\langle z, z \rangle}$, with the following properties:

- $\|z\| \geq 0$, and $\|z\| = 0$ iff $z = 0$.
- $\|\alpha z\| = |\alpha| \|z\|$ for any scalar $\alpha$.
- $\|z + w\| \leq \|z\| + \|w\|$ (triangle inequality).

### B.3 METRIC

The inner-product norm induces a metric $d$ on $\mathcal{Z}$: $d(z, z') = \| z - z' \| = \sqrt{\langle z - z', z - z' \rangle}$, which satisfies:

- $d(z, z') \geq 0$, and $d(z, z') = 0 \iff z = z'$.
- $d(z, z') = d(z', z)$ (symmetry).
- $d(z, z') \leq d(z, z'') + d(z'', z')$ (triangle inequality).

Introducing $d$ is necessary because **First**, it provides a principled way to compare embeddings: $d(\phi(s), \phi(g))$ quantifies the similarity of states. **Second**, many representation-learning objectives (e.g. contrastive or temporal losses) are formulated in terms of distances. **Besides**, Hilbert-space policies use inner products or distances to define intrinsic rewards, ensuring temporally coherent behaviors. This geometric structure underpins both zero-shot inference and goal-conditioned planning in our foundation policy.

## C  DERIVATION OF CLOSED FORM SOLUTIONS

In order to solve for the optimal vector $z$ in the latent space $\mathcal{Z}$ to get the near-optimal policy solutions. The objective is to minimize the difference between the observed reward and the linear combination of the feature matrix and $z$. Formally, the optimization problem is:

$$z^* = \arg \min_{z \in \mathcal{Z}} \mathbb{E}_D \left[ (r(s, a, s') - \langle \phi(s, a, s'), z \rangle)^2 \right] \tag{8}$$

where the observed reward for transition $(s, a, s')$ is denoted as $r(s, a, s')$, the feature vector describing the transition is $\phi(s, a, s') \in \mathbb{R}^d$, the task-specific latent vector to optimize is $z \in \mathbb{R}^d$, the expectation over the dataset $D$ is represented by $\mathbb{E}_D$, and the inner product in $\mathbb{R}^d$ is denoted by $\langle \cdot, \cdot \rangle$.

In order to get the closed-form solution for this optimization problem, we have the derivation as follows. We can simplify the goal in matrix form for $n$ samples in the dataset:

$$\mathcal{L}(z) = (r(s, a, s') - \langle \phi(s, a, s'), z \rangle)^2 = \| r - \Phi z \|_2^2 \tag{9}$$

where the vector of observed rewards is $r = [r_1, r_2, \ldots, r_n]^\top$ with dimensions $n \times 1$, the feature matrix is $\Phi = [\phi_1, \phi_2, \ldots, \phi_n]^\top$ with dimensions $n \times d$, and the latent vector to optimize is $z \in \mathbb{R}^d$.

Expanding the loss:

$$\| r - \Phi z \|_2^2 = r^\top r - 2r^\top \Phi z + z^\top \Phi^\top \Phi z. \tag{10}$$

And then differentiating with respect to $z$:

$$\frac{\partial \mathcal{L}(z)}{\partial z} = -2\Phi^\top r + 2\Phi^\top \Phi z. \tag{11}$$

Because the unique global minimum exists if the function is differentiable and the second derivative is positive semidefinite, and the critical point where the derivative equals zero corresponds to the global minimum, we next set the derivative to zero:

$$\begin{aligned} -2\Phi^\top r + 2\Phi^\top \Phi z &= 0, \\ \Phi^\top \Phi z &= \Phi^\top r. \end{aligned} \tag{12}$$

Then we rearrange the result, which gives the closed-form solution:

$$z^* = (\Phi^\top \Phi)^{-1} \Phi^\top r \tag{13}$$

where $\Phi^\top \Phi$ is the covariance matrix of the features, $\Phi^\top r$ represents the correlation between features and rewards, $z^*$ minimizes the squared error between the observed rewards and the predictions given by $\Phi z$.

## D  NOTATION SUMMARY

In this section, we provide a table of detailed notation and explanations that appear in the paper. It provides a comprehensive summarization of the terms used in the preliminary and methodology sections.

| Symbol | Description |
|---|---|
| $\mathcal{M} = (\mathcal{S}, \mathcal{A}, P, r, \mu, \gamma)$ | Markov decision process. |
| $\pi(a|s)$ | Policy distribution over actions. |
| $\varphi(s)$ | State encoder to latent space. |
| $\mathcal{Z} = \mathbb{R}^D$ | Hilbert latent space. |
| $d(s, g) = \|\varphi(s) - \varphi(g)\|$ | Latent distance between states. |
| $\pi(a|s, z)$ | Latent-conditioned foundation policy. |
| $r(s, z, s') = \langle \varphi(s') - \varphi(s), z \rangle$ | Intrinsic reward from latent direction. |
| $z^*$ | Task-specific optimal latent vector. |
| $\Phi \in \mathbb{R}^{N \times D}$ | Feature matrix of transitions. |
| $r \in \mathbb{R}^N$ | Reward vector. |
| $z = (\Phi^\top \Phi)^{-1} \Phi^\top r$ | Least-squares latent solution. |
| $\Delta P = P_{\text{real}} - P_{\text{sim}}$ | Dynamics discrepancy. |
| $\Phi_{\text{real}} = \Phi_{\text{sim}} + \Delta \Phi$ | Real feature matrix. |
| $z_{\text{src}}$ | Latent from joint regression. |
| $\lambda$ | Weight for real-domain samples. |
| $\eta = \text{MetaDynamic}(\{\varphi(\tilde{s}_j)\})$ | Permutation-invariant dynamics signature. |
| $g_\theta(z_{\text{src}}, \eta)$ | Adapter network for latent refinement. |
| $z_{\text{final}}$ | Refined latent for deployment. |

Table 2: Notation summary for the proposed latent adaptation framework.

| Setting Name | Parameter Changes | Description |
|---|---|---|
| Default | Default | default setting of $E_{\text{sim}}$ |
| G1 | Gravity $-9.8 \rightarrow -15$ | Gravity-level 1 $E_{\text{real}}$ – easiest |
| G2 | Gravity $-9.8 \rightarrow -24$ | Gravity-level 2 $E_{\text{real}}$ – middle easy |
| G3 | Gravity $-9.8 \rightarrow -34$ | Gravity-level 3 $E_{\text{real}}$ – middle hard |
| G4 | Gravity $-9.8 \rightarrow -44$ | Gravity-level 4 $E_{\text{real}}$ – hardest |
| F1 | Friction $\rightarrow [4, 0.4, 0.4]$ | Friction-level 1 $E_{\text{real}}$ – easiest |
| F2 | Friction $\rightarrow [5, 0.5, 0.5]$ | Friction-level 2 $E_{\text{real}}$ – slight easy |
| F3 | Friction $\rightarrow [6, 0.6, 0.6]$ | Friction-level 3 $E_{\text{real}}$ – middle easy |
| F4 | Friction $\rightarrow [7, 0.7, 0.7]$ | Friction-level 4 $E_{\text{real}}$ – slight hard |
| F5 | Friction $\rightarrow [8, 0.8, 0.8]$ | Friction-level 5 $E_{\text{real}}$ – harder |
| F6 | Friction $\rightarrow [18, 1.8, 1.8]$ | Friction-level 6 $E_{\text{real}}$ – hardest |

Table 3: Simulator-to-Real Configurations for $E_{\text{real}}$

## E  MORE EXPERIMENTAL RESULTS

### E.1  DETAILS OF EXPERIMENTAL SETUPS

**(1). Computing Resources**: All case studies are conducted on a local workstation equipped with an NVIDIA GeForce RTX 4090 GPU (24 GB memory), CUDA 12.2, and driver version 535.183.01,

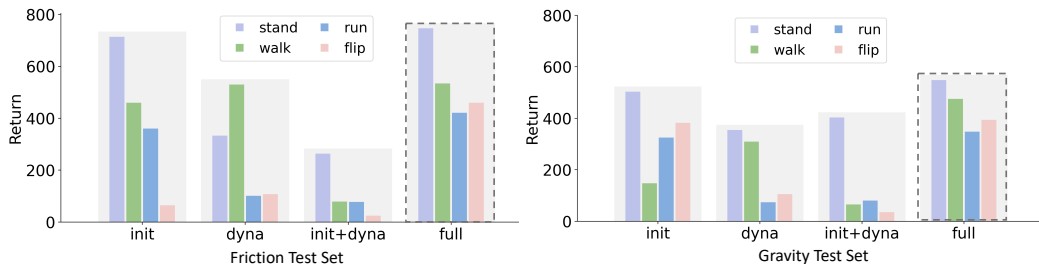

Figure 8: The full ablation study results. The left side shows the experiment performed on the friction test set, and the right side shows the experiment result on the gravity-based test cases. As shown in the dotted line box, the full model performs the best among the other three ablated model structures.

running under Ubuntu Linux. The main experiments reported in the tables were conducted with a high-performance computing cluster equipped with NVIDIA A100 GPUs (80 GB memory).

**(2). Training and Data Collection Configs**: All the pre-training of the foundation models was executed with 2,000,000 episodes, we adopt a unified configuration across all environments following a Hydra-based setup (Yadan, 2025). We employ the DDPG agent pre-training and state-based observations (with optional pixel-based experiments like `Kitchen` using frame stacking and action repeat). In the total episodes, we set the first 1,000 frames for seeding the replay buffer. Replay buffer capacity is fixed at 1,000,000 transitions, serve as the source-domain dataset $D_{src}$. In addition, we also construct a target-domain dataset $D_{tar}$ by executing the pre-trained policy in the target environment for 5,000 rounds of interaction.

**(3). Baseline Implementation Details**: In this section, we provide details of the baseline models' training. For PAD (Hansen et al.) model, we follow the official repository as released in the project website [3] Besides, since the grounded action transformation methods could provide adaptation on the policy network in real time by training a forward model and inverse model, and ground the action based on the estimated real-world system dynamics, thus, we adapt this branch of method and conduct online adaptation, the implementation mainly follow the release code base at public repository [4]. To make a fair comparison in the online adaptation time, we fixed the epochs of adaptation as 200 across the main table's experiments. In uncertainty-aware GAT (UGAT), it integrates the evidential deep learning module for uncertainty prediction in the inverse module, and the uncertainty will be used to decide whether to accept new proposed action (low-uncertainty) or take the original action (high-uncertainty), this method generally provides consistent improvement than Vanilla-GAT model based on our experiment.

### E.2 INTERPRETATION OF THE DATA QUALITY RESULTS.

Following the discussion in Sec. 5.1 (4), Fig. 9 (four panels) extends the analysis from *stand* to *walk*, *run*, and *flip* under three corruption modes (`drop`, `mask`, `noise`). Two patterns are consistent across tasks. First, `drop` exhibits *graceful degradation*: at less than 50% removal rate, *stand* and *walk* decline only mildly, indicating that reducing target-domain sample size might weaken the adapter's evidence, but our method could still perform relatively robustly. In contrast, *run* and *flip* deteriorate more sharply at the same rate; these skills start from a lower baseline because we are *zero-shot prompted to a new task while adapting to new environments*, making them more sensitive when losing the target domain data and amplifying variance in the latent refinement.

Second, `mask` *fails early* in four tasks, which shows that, converting selected transitions to zeros injects biased signals into the squared-loss objective of Eq. 5, pulling the refined latent toward degenerate regions and causing rapid collapse once a moderate fraction is masked. By comparison, `noise` shows a *nonlinear* response: light corruption can act like a mild regularizer (occasionally

---

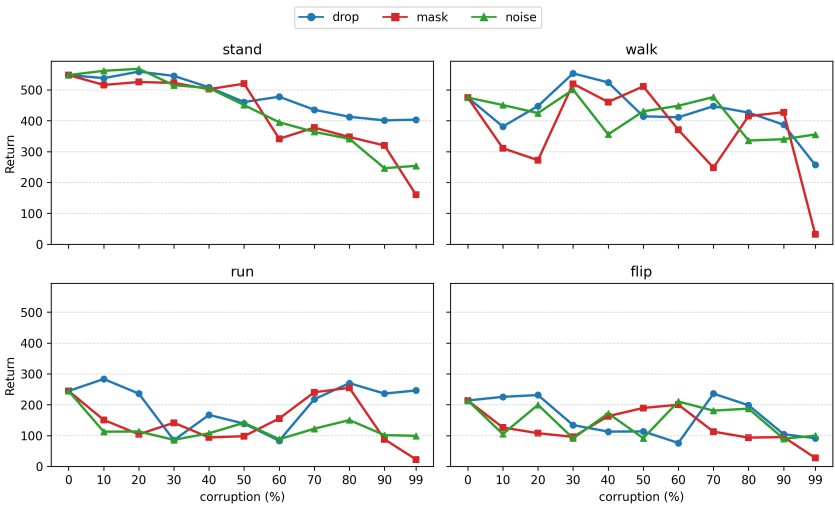

Figure 9: More comprehensive data quality analysis on four different tasks with three modes of corruptions and varied corruption rates.

producing small bumps), but heavier noise degrades performance, particularly for the dynamics-sensitive *run/flip*, because it disrupts the consistency between $(s, s')$ on which the adapter relies.

Overall, the curves suggest a task-wise sensitivity hierarchy: *stand* ≈ *walk* are comparatively robust to sparsity, whereas *run* and *flip* are fragile both to reduced evidence (50% `drop`) and to signal corruption (`mask`/`noise`). Taken together, these results highlight that **Found-adapt** is *surprisingly resilient*: it delivers stable performance under substantial sparsity (e.g., ≤50% `drop` in *stand*/*walk*) and degrades gracefully relative to more brittle perturbations. *However*, on the more demanding skills: *run* and *flip*, we observe that, the curves exhibit noticeably larger variance under comparable corruption. We attribute this to the compounded difficulty of *zero-shot adaptation to a new task while adapting*: contact- and timing-sensitive dynamics amplify small inconsistencies between $(s, s')$, and the induced latent landscape can be more rugged. In conclusion, **Found-adapt** still delivers consistent sim-to-real lift on unseen tasks by efficient parameter tuning, without retraining the whole policy, which goes beyond conventional pipelines.

## F  FEASIBILITY IN GOAL-CONDITIONED RL SETTING.

We show that **Found-adapt** applies to goal-conditioned RL (GCRL) with no change to the foundation policy or training loop, only to how the task latent is instantiated from a goal. Let $g$ denote a goal (e.g., a target state or target representation). As in the main text, the foundation policy is $\pi(a|s, z)$, the transition feature is $\phi(s, a, s') \in \mathbb{R}^D$, and rewards are modeled linearly in the latent:

$$r(s, a, s') \approx \langle \phi(s, a, s'), z \rangle.$$

For GCRL we simply index the latent by the goal and write $z_g$. Many standard goal rewards (e.g., indicator of reaching $g$ or a shaped distance to $g$) admit a linear fit in the same feature space; we denote their labels for transition $j$ by $\tilde{r}_{g,j}$.

**Goal-to-latent mapping in the simulator.**  Given a set of simulator transitions $\{(s_i, a_i, s_i')\}_{i=1}^N$ and a goal $g$, form

$$\Phi_{\text{sim}} \in \mathbb{R}^{N \times D} \quad \text{with rows} \quad \Phi_{\text{sim},i}^\top = \phi(s_i, a_i, s_i'), \qquad \tilde{r}_{g,\text{sim}} \in \mathbb{R}^N.$$

The goal-specific source latent $z_g^{\text{src}}$ is obtained exactly as in the main method (weighted/regularized least-squares in the same feature basis):

$$z_g^{\text{src}} = \arg\min_{z \in \mathbb{R}^D} \frac{1}{N} \left\| \Phi_{\text{sim}} z - \tilde{r}_{g,\text{sim}} \right\|_2^2 + \lambda \|z\|_2^2 = \left( \Phi_{\text{sim}}^\top \Phi_{\text{sim}} + \lambda I \right)^{-1} \Phi_{\text{sim}}^\top \tilde{r}_{g,\text{sim}}.$$

When partial target data are also available, the same stacked/weighted normal equations used in the main method can be applied to include them.

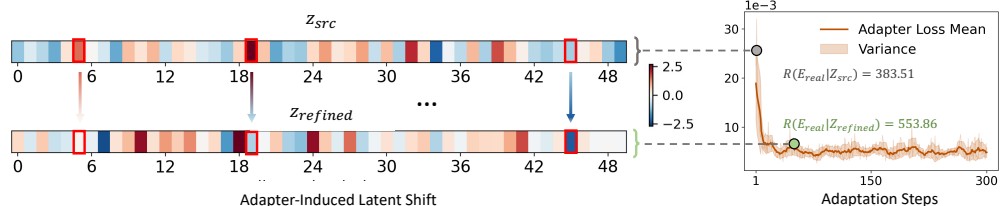

Figure 10: The visualization of the $z_{\text{src}}$ and $z_{\text{refined}}$ (after normalization). The left part, from $z_{\text{src}}$ to $z_{\text{refined}}$, it shows the latent space changes during the adaptation process. Together with the loss function decrease, the latent representation is changing, which leads to the final better solution in the $E_{real}$ domain dynamics, yielding better performance in the $E_{real}$ rewards. This visualization also confirms the refinement process is smooth and stable, demonstrating that the lightweight adapter does not introduce instability. Due to the simple network architecture, it has a low risk of overfitting.

**Target-domain refinement via the latent adapter.** At inference, **Found-adapt** already computes an environment signature $\eta$ from target transitions (DynamicNet over $\phi$-features) and refines the latent via the adapter $g_\theta(\cdot, \cdot)$. For a given goal $g$, define the target design matrix and labels

$$\Phi_{\text{tar}} \in \mathbb{R}^{M \times D}, \qquad \tilde{r}_{g,\text{tar}} \in \mathbb{R}^M,$$

constructed from the *same* target transitions but goal-relabelled rewards (e.g., success indicator for $g$, or a shaped distance to $g$ in your representation). The adapter objective used throughout the paper directly applies:

$$\mathcal{L}_{\text{tar}}(\theta; g) \;=\; \frac{1}{M} \sum_{j=1}^{M} \left\| \Phi_{\text{tar},j}\, g_\theta\!\left(z_g^{\text{src}}, \eta\right) - \tilde{r}_{g,j} \right\|_2^2, \qquad z_g^{\text{final}} \;=\; \sqrt{D}\, \frac{g_{\theta^*}\!\left(z_g^{\text{src}}, \eta\right)}{\left\| g_{\theta^*}\!\left(z_g^{\text{src}}, \eta\right) \right\|_2}.$$

The resulting $z_g^{\text{final}}$ conditions the *same* foundation policy $\pi(a|s, z)$ for the goal $g$, i.e., $\pi(a|s, z_g^{\text{final}})$, without retraining the policy.

Because $g$ only changes the scalar labels $\tilde{r}_{g,.}$ while keeping the feature design matrices $\Phi_{\text{sim}}$ and $\Phi_{\text{tar}}$ *unchanged*, **Found-adapt** is compatible with standard GCRL data strategies: (i) *hindsight relabelling*, for any stored transition $(s, a, s')$, reuse it for many $g$ by recomputing $\tilde{r}_g$; and (ii) *on-the-fly goal queries*, compute $z_g^{\text{src}}$ (closed form) and its one-step refinement $z_g^{\text{final}}$ per goal at inference. No changes to $\phi$, $\eta$, or $\pi$ are needed.

**Special case: representation-shaped goals.** If the goal reward is a distance in the same representation used by $\phi$, e.g.,

$$\tilde{r}_g(s, a, s') \;\propto\; -\left\| \varphi(s') - \varphi(g) \right\|_2^2,$$

then, under the linear reward model already assumed by **Found-adapt**, the induced $z_g$ coincides with the least-squares projection of this goal signal onto the feature span of $\phi(s, a, s')$. Hence the same identifiability and conditioning arguments in the main method carry over, and the adapter uses $\eta$ to correct residual simulation-to-real mismatch *per goal*.

Thus, GCRL in **Found-adapt** amounts to swapping task IDs for goals when forming the reward labels $\tilde{r}_{g,.}$; all estimators and refinements, least-squares for $z^{\text{src}}$, DynamicNet environment signature $\eta$, and the adapter $g_\theta$ remain unchanged. Thus, foundation policy $\pi(a|s, z)$ can be potentially deployed across *many* goals by computing $z_g^{\text{final}}$ at inference, preserving the inference-time adaptation advantages demonstrated in the main (task-conditioned) setting.

# G   DETAILED MODEL CONFIGURATIONS

The MetaDynamic network used in Sec. 4.2.B is a permutation-invariant set encoder. Given a set of target-domain encoded states $\{\phi(\tilde{s}_j)\}_{j=1}^N$, it produces a dynamics signature $\eta \in \mathbb{R}^K$ through an elementwise transformation, pooling, and a global projector such as below:

$$\eta = \rho\left( \frac{1}{N} \sum_{j=1}^{N} \phi_{\text{enc}}(\phi(\tilde{s}_j)) \right) \tag{14}$$

**Architecture:** We use a lightweight DeepSet-style module as below:

- elementwise encoder $\phi_{\text{enc}}$: $\text{MLP}(d \to 128 \to 128, \text{ ReLU})$,
- permutation-invariant mean pooling,
- projector $\rho$: $\text{MLP}(128 \to 128 \to K, \text{ ReLU})$ with $K = 64$.

**Pre-training:** The encoder is trained offline using simulator trajectories collected under 12 dynamics variations (gravity, friction, and actuator-strength perturbations). Training uses a combined loss:

$$\mathcal{L} = \mathcal{L}_{\text{CE}} + \alpha\, \mathcal{L}_{\text{NTXent}}, \quad \alpha = 0.1 \tag{15}$$

The cross-entropy term $\mathcal{L}_{\text{CE}}$ encourages the encoder to separate different dynamics regimes, while the NT-Xent loss $\mathcal{L}_{\text{NTXent}}$ (Chen et al., 2020) pulls together sets from the same dynamics and pushes apart sets from different ones, producing a smooth and discriminative dynamics signature.

**Deployment:** After pre-training, the parameters of MetaDynamic are frozen. During deployment, it receives a small set of real-domain state encodings and outputs the corresponding dynamics signature $\eta$ without further updates.

## H  DISCUSSION ON FUTURE WORK

**Regularization for stable latent refinement.**  The refinement of the latent vector through the adapter $g_\theta$ is supervised by the projection $\|\Phi_{\text{targ}}\, g_\theta(z_{\text{src}}, \eta) - r_{\text{targ}}\|^2$ which constrains the solution only through the row space of $\Phi_{\text{targ}}$. In theory, multiple latent vectors may achieve similar projection error, and this can potentially introduce a certain degree of overfitting. We find this effect to be limited when the latent dimension is small, and the adapter is conditioned on $z_{\text{src}}$, which serves as a strong prior and restricts the refinement to a meaningful neighborhood of the initial solution. Moreover, the adapter is a low-capacity MLP that is optimized for only a few hundred steps, providing a strong form of implicit regularization. Adding an appropriate regularization term toward $z_{\text{src}}$ or incorporating geometric constraints in the latent space may further enhance the refinement, which eventually provides better identifiability and improved performance under stronger perturbations.

**Extending to real-world sim-to-real adaptation.**  Our experiments follow the widely adopted sim-to-sim evaluation protocol, which provides controlled and repeatable dynamics variations for isolating the effect of latent-space adaptation. This setting is both standard and practical for analyzing dynamic generalization. An important direction for future work is to deploy the proposed method in real hardware environments to more directly assess its sim-to-real capability. Additionally, replacing state-based features with image observations or vision-based encoders would make the pipeline more applicable to realistic robotic systems, where dynamics shifts often manifest through visual changes. Such extensions would further validate the flexibility of foundation-policy adaptation and broaden its applicability to perceptual and real-world domains.

