# OpenReview forum: "Latent Adaptation of Foundation Policies for Sim-to-Real Transfer"
_ICLR.cc/2026/Conference — ICLR 2026 Poster_

### Official Review · Reviewer_A9Lz · 2025-10-16

**Soundness:** 3
**Presentation:** 3
**Contribution:** 3
**Rating:** 8
**Confidence:** 2

**Summary:**

This paper introduces Found-adapt, a framework for efficient sim-to-real transfer in reinforcement learning that adapts pretrained foundation policies through latent-space alignment rather than policy retraining. The method first trains a foundation policy on offline simulator trajectories in a Hilbert-space embedding that encodes reusable behavioral primitives. At deployment, adaptation is performed in three lightweight stages: a cross-domain least-squares latent alignment that integrates simulator and limited real data, a MetaDynamic network that extracts a permutation-invariant dynamics signature from the target domain, and a signature-guided adapter that refines the latent vector to align behavior with new dynamics. This parameter-efficient approach enables adaptation without modifying policy weights. Extensive experiments on locomotion tasks (Walker, Cheetah, Quadruped, AntMaze, Kitchen) under varying gravity and friction demonstrate that Found-adapt substantially narrows the sim-to-real performance gap compared to strong baselines (GAT, UGAT, PAD), achieving similar performance with orders-of-magnitude lower adaptation time. Detailed ablations, sensitivity analyses on data corruption, and studies linking adaptation loss to performance further support the framework’s robustness and practical potential for real-world RL deployment.

**Strengths:**

1. The paper reframes sim-to-real transfer as latent alignment rather than policy re-optimization. This perspective—adapting a frozen foundation policy by operating in its latent space—is conceptually fresh and could generalize across domains and tasks. The Hilbert-space foundation policy formulation (Sec. 4.1) gives a principled way to tie skill structure to temporal dynamics, aligning well with recent trends in foundation models for control.

2. The method is well-structured and mathematically grounded. Equations (1–6) and Figure 2 outline a coherent pipeline linking the least-squares latent solution, MetaDynamic signature, and adapter update. The combination of permutation-invariant dynamics encoding and parameter-efficient adaptation is elegant and interpretable.

3. Experiments span multiple environments and perturbations (gravity G1–G4, friction F1–F6), with comparisons to both task-specific (PAD) and dynamics-specific (GAT/UGAT) baselines (Table 1, Figures 3–9). Found-adapt consistently ranks among the top methods, often within 5-10% of PAD but with orders-of-magnitude faster adaptation (seconds vs. hours).

4. The ablation study (Figure 4a) carefully isolates the contribution of the initial alignment, dynamics signature, and adapter. The sensitivity analyses on data corruption (Figures 10–11) and the strong correlation between adaptation loss and performance improvement (Figure 6) demonstrate a rare level of diagnostic rigor for sim-to-real studies.

5. The approach avoids retraining the policy or encoder entirely, performing adaptation through small-scale optimization over a latent vector. This is appealing for real-world robotics, where data collection is costly. The inclusion of analysis on robustness to noisy and sparse target-domain data (Sec. 5.1 (4)) adds credibility to practical deployability.

**Weaknesses:**

1. Dynamics signature lacks true transition information

The MetaDynamic encoder (Sec. 4.2B) computes η solely from unordered states, ignoring actions and successor-state relations. As a result, η summarizes state visitation statistics rather than transition dynamics P(s′|s,a). This makes η non-identifiable and may cause adaptation to correct policy-induced distribution shifts rather than genuine system-dynamics changes.

2. Latent update relies on unstable closed-form inversion

The least-squares solutions (Eqs. 2-3) require inverting ΦᵀΦ and AᵀA without any regularization or conditioning safeguards. Appendix C provides no analysis of rank deficiency or noise sensitivity. In low-data or ill-conditioned regimes, this can yield unstable latent estimates, undermining the method’s reliability under realistic sim-to-real conditions.

3. Adapter optimization is underconstrained

The adapter gθ is trained with a simple projection loss (Eq. 5), which constrains the latent only within the row space of Φ_tar. Without explicit regularization or geometric constraints toward z_src, multiple gθ can fit the same projections, causing overfitting and inconsistent behavior in closed-loop execution.

4. Target reward requirement contradicts the “unlabeled” claim

Although the paper emphasizes adaptation from limited unlabeled data, both Eq. 3 and Eq. 5 require real-domain reward labels. This assumption is not addressed or justified, and in many real-world settings such rewards are unavailable or sparse. The omission makes the “label-efficient” narrative misleading and limits practical applicability.

**Questions:**

please address the concerns above

---

> ### Author Response · Authors · 2025-11-26
> **Response to Reviewer A9Lz**
>
> **We appreciate the reviewer A9Lz for the recognition of our work, we are more than willing to provide more explanations to the raised questions**.
>
> > W1: Dynamics signature lacks true transition information
>
> 1. The goal of the MetaDynamic network is to extract a hidden descriptor for the underlying system physics, as stated in line 264: `a compact descriptor encoding ...` Such a descriptor is intended to be independent of individual transitions and instead reflect system-level properties, which is why we specifically design a permutation-invariant network for this purpose.
>
> 2. We indeed capture transition information through the regression objective $\Phi z \approx r$, where $\Phi$ contains Hilbert-embedded successor-state features. Thus, $\eta$ serves as a dynamics context that modulates the latent refinement.
>
> 3. Recovering the full transition kernel $P(s' \mid s,a)$ would move toward model-based RL, which aims to identify system dynamics from transition probabilities, this objective is not quite feasible under the work's limited-data adaptation setting.
>
>
>
> > W2: Latent update relies on unstable closed-form inversion
>
> Thanks for the question.
>
> 1. We note that the solution in Eq. (2–3) is well-conditioned in both our setting and the traditional HILP work[1]. First, the latent dimension is small (e.g., $D=32$), so the matrix $A^\top A$ is only $32\times 32$, making the system solving process numerically stable. Moreover, $\Phi$ and $A$ contain Hilbert-embedded successor-state features, which are high-dimensional and not simple that possibly lead to degenerate, so rank deficiency is very unlikely in practice.
>
> 2. We do not invert matrices explicitly; we use the standard linear solver in PyTorch using `torch.linalg.lstsq(A, b)`, which internally performs a stable decomposition (QR/SVD). This avoids the numerical issues typically associated with direct matrix inversion.
>
> 3. The downstream adapter (Fig. 5) reflects that the stability of the least-squares solution under corrupted situations like drop, mask, or noise: Fig. 5(a) shows the method can achieve good returns under high corruption rates. Since the refinement operates on the least-squares output $z_{\text{src}}$, such stability would not occur if the initial solution were numerically unstable. This provides empirical support that the weighted least-squares step provides stable and reasonable solutions in the sim-to-real domains we consider.
>
>
> *[1] Park, S., Kreiman, T. &amp; Levine, S.. (2024). Foundation Policies with Hilbert Representations. Proceedings of the 41st International Conference on Machine Learning, in Proceedings of Machine Learning Research 235:39737-39761.*
>
> > W3: Adapter optimization is underconstrained
>
> We appreciate the reviewer’s observation. In principle, optimizing $g_\theta$ with the projection loss $\|\Phi_{\text{targ}}\, g_\theta(z_{\text{src}},\eta)-r_{\text{targ}}\|^2$ can admit multiple latent vectors that achieve similar projection error. We acknowledge that this could potentially introduce a certain degree of inevitable overfitting.
>
> However, in practice, this effect is limited for two reasons. First, the latent dimension is very small, and the adapter receives $z_{\text{src}}$ as a strong prior, which restricts the refinement to a small and meaningful region of the latent space. Second, the adapter has low capacity and is optimized for only hundreds of steps (in our setting), which provides implicit regularization.
>
> We agree that introducing an appropriate regularizer toward $z_{\text{src}}$ could control and mitigate the overfitting, further strengthen the identifiability of the refinement, and may help improve the results in Fig. 3. We have added this discussion in the revised version of Section I for future work. Thanks for the constructive thoughts.
>
> > W4: Target reward requirement contradicts the “unlabeled” claim
>
> In our paper, the term “unlabeled offline data” (line 19, 110, 162) refers specifically to the data used in *pretraining the foundation policy*. This pretraining dataset is reward-free and unlabeled in the standard sense, which makes it easy to collect and rich in information.
>
> For the adaptation stage, we use only a small amount of target-domain interaction data together with the environment-provided reward. It is used solely to calibrate the latent and bridge the dynamics gap, not to provide task supervision.
>
> Thus, the reward requirement `does not contradict` the “unlabeled” claim for pretraining the foundation policy.
>
>
>
> **We sincerely appreciate the thoughtful insights from reviewer A9Lz. We have included the suggested revisions in the current version of the PDF, and we are happy to provide more clarifications if needed. Thank you for your time and effort.**

---

### Official Review · Reviewer_NFbW · 2025-10-16

**Soundness:** 3
**Presentation:** 2
**Contribution:** 2
**Rating:** 4
**Confidence:** 3

**Summary:**

This paper introduces Found-adapt, a novel framework for addressing the sim-to-real transfer challenge in reinforcement learning. The key contribution is a latent space adaptation method that enables foundation policies to transfer across domains without expensive policy retraining. The approach consists of three main components: (1) cross-domain initial alignment via weighted least-squares regression, (2) a permutation-invariant MetaDynamic network that extracts dynamics signatures, and (3) a signature-guided adapter that refines latent representations. The method is evaluated on multiple locomotion tasks under varying system dynamics (gravity and friction variations), demonstrating effective sim-to-real gap mitigation while maintaining computational efficiency.

**Strengths:**

1. Clear problem formulation: The paper addresses a critical deployment challenge where traditional sim-to-real methods require expensive retraining for each new environment, and motivates the solution through an intuitive biological analogy of how humans adapt existing motor skills to new conditions without relearning from scratch.

2. Rigorous theoretical foundation: The approach is built on solid mathematical principles with formal analysis of how environment changes affect policy performance, closed-form solutions with clear assumptions, and well-justified design choices where each component addresses specific aspects of the adaptation problem.

3. Exceptional computational efficiency: The method achieves adaptation in seconds compared to hours required by baseline approaches, representing orders of magnitude speedup by freezing most network parameters and performing lightweight updates only in the latent space.

**Weaknesses:**

1. Limited domain coverage: The experimental validation is restricted exclusively to locomotion tasks, limiting the claimed applicability as a general foundation policy framework.

2. Sim-to-sim evaluation only: All experiments use simulated environments with modified physics parameters rather than actual real-world robots, raising concerns about whether the method can handle the full complexity of real-world dynamics.

3. Limited baseline comparisons: The paper primarily compares against Direct-Transfer, GAT/UGAT, and PAD while missing comparisons with recent domain randomization methods, meta-RL approaches designed for adaptation, and other foundation model techniques, which weakens the empirical validation of the claimed advantages.

**Questions:**

1. Scalability to high-dimensional action spaces: All tested environments have relatively low-dimensional continuous action spaces - how does the method scale to high-dimensional or discrete action spaces, and are there theoretical or practical limits to the dimensionality it can handle?

2. Comparison of adaptation mechanisms: How does your latent space adaptation compare conceptually and empirically to other lightweight adaptation strategies such as context-based meta-learning, fine-tuning only the final policy layers, or dynamic rescaling of actions? What makes latent adaptation specifically advantageous for the sim-to-real problem?

---

> ### Author Response · Authors · 2025-11-27
> **Response to Reviewer NFbW (1/2)**
>
> **Thank reviewer NFbW for taking time to review our work and provide valuable feedback. We would like to explain more on the raised questions (Q) and weaknesses (W).**
>
> ## Questions
>
> > Q1: Scalability to high-dimensional action spaces
>
> Thanks for the question. **Since this work focuses on the domain adaptation, we would like to clarify that the proposed method is not tied to the dimensionality of the action space**. This is because the adaptation operates in the latent space of the foundation policy. As shown in Eq. 3 and Eq. 5, the weighted least-squares step uses Hilbert-embedded successor-state features $\Phi(s,s')$ and predicts rewards, so its dimensionality depends only on the latent dimension $D$, not constrained by action dimension. Similarly, the MetaDynamic module and the adapter $g_\theta$ refine the latent vector $z$ and are also independent of the size or type of the action space.
>
> Thus, scaling of the action space affects the expressiveness of the foundation policy, but **does not affect the proposed adaptation mechanism** proposed in this work.
>
>
> > Q2: Comparison of adaptation mechanisms.
>
> We would like to discuss the differences of these methods compared to ours. **`However, due to the lack of specific references, it is hard for us to identify concrete baselines to re-implement for empirical comparison. We would be very happy to further expand our analysis if the reviewer might kindly point us to specific papers that apply alternative strategies to sim-to-real adaptation. Thank you very much!`**
>
> We conducted a more comprehensive survey following the reviewer's general guide, and based on the literature we found (relevant ideas), we provide a conceptual discussion regarding this question.
>
> 1. Compare to context-based meta-learning [1]:
>
> Regarding context-based meta-learning, a representative work is MAML [1], but it is designed for rapid `task adaptation` instead of handling `dynamics shifts or sim-to-real transfer`. MAML assumes fixed environment dynamics and meta-learns an initialization that can be quickly fine-tuned to solve new reward functions or behavioral goals. **In contrast**,  our setting keeps the task fixed and focuses on adapting to changes in system dynamics  (e.g., gravity, friction, etc). Thus, even though context meta-learning offers conceptual insight, it is not directly applicable to sim-to-real adaptation, which requires reasoning about system parameter changes rather than task variation. Our latent-space adaptation targets dynamics-induced representation drift and is specifically designed for sim-to-real scenarios.
>
> *[1] Finn, Chelsea, Pieter Abbeel, and Sergey Levine. "Model-agnostic meta-learning for fast adaptation of deep networks." International conference on machine learning. PMLR, 2017.*
>
>
> 2. Compare to fine-tuning only the final policy layers of the foundation policy [2] [3]:
>
> If to adapt the foundation policy by fine-tuning only its final layers, the adaptation would still require gradient updates on target(real)-domain trajectories and would modify a subset of the policy parameters. This brings two challenges under the sim-to-real setting: (i) parameter-space finetuning is inherently sensitive to reward noise and can easily destabilize a pretrained policy, especially when only `a small amount of real data` is available; and (ii) fine-tuning is task-specific, the different behaviors (e.g., walk, flip, etc.) require different finetuning data and returns, so the resulting adaptation cannot be shared across task.
>
> **Instead** of changing policy parameters, we refine only a compact latent vector that governs how the policy interprets the dynamics, then our method learns the latent system shifts and forms a policy by data prompting. Such task-agnostic way ensures a stable adaptation: no online gradients, the update is lightweight and consistent across tasks.
>
> *[2] Yin, Patrick, et al. "Rapidly Adapting Policies to the Real World via Simulation-Guided Fine-Tuning." arXiv preprint arXiv:2502.02705 (2025).*
>
> *[3] Bharadhwaj, Homanga, et al. "A data-efficient framework for training and sim-to-real transfer of navigation policies." 2019 International Conference on Robotics and Automation (ICRA). IEEE, 2019.*

---

> ### Author Response · Authors · 2025-11-27
> **Response to Reviewer NFbW (2/2)**
>
> 3. Compare to the dynamic rescaling of actions:
> There are recent work that incorporates action-modulation or residual correction for sim-to-real. These methods adjust torque magnitudes or action outputs to compensate for the mismatch in actuation or contact parameters like [4].
>
> This branch of methods typically operates directly in the action space by scaling, biasing, or adding a residual term to the torque commands. Even though this allows quick compensation for specific system differences, but the adaptation remains highly local, task-specific, and constrained to the structure of the action space. This local correction, while effective, requires accurate system identification to decide the compensation term/equation.
>
> In our method, the refinement of a latent vector $z$ directly identifies the system abstraction and identifies the gap. This provides a task-agnostic, representation-level correction that generalizes across behaviors (walk, stand, flip, run) and captures domain-level changes instead of action-level biases. This adaptation is more systematic, lightweight, and avoids the brittleness associated with torque-space modification.
>
> *[4] Luo, Zeren, et al. "Moral: Learning morphologically adaptive locomotion controller for quadrupedal robots on challenging terrains." IEEE Robotics and Automation Letters 9.5 (2024): 4019-4026.*
>
>
>
> ## Weaknesses
>
> > W1: Limited domain coverage
>
> Since our focus is on the sim-to-real adaptation technique rather than the task breadth of foundation policy, thus, we build upon the backbone Hilbert representation foundation model, which is already well-acknowledged that suits multiple control domains.
>
> Locomotion tasks are chosen because they allow controlled and repeatable manipulation of dynamics parameters (gravity, friction, etc), which is practical for assessing sim-to-real adaptation. Since our method adapts only the latent representation and does
> not rely on task-specific structures, it is orthogonal to the choice of task and can be applied to other domains supported by the foundation policy. We will also clarify this in the revision.
>
> > W2: Sim-to-sim evaluation
>
> Thanks for the concern. In order to realize controlled and repeatable variations in system dynamics for evaluation, we follow a widely-adopted sim-to-sim protocol as in work [5][6][7], to verify the sim-to-real performance. We have included the discussion and potential future work for sim-to-real evaluation in the paper Section I, part 2.
>
> *[5] Eysenbach, B., Chaudhari, S., Asawa, S., Levine, S., & Salakhutdinov, R. Off-dynamics reinforcement learning: Training for transfer with domain classifiers. In International Conference on Learning Representations (ICLR).*
>
> *[6] Hansen, Nicklas, and Xiaolong Wang. "Generalization in reinforcement learning by soft data augmentation." 2021 IEEE International Conference on Robotics and Automation (ICRA). IEEE, 2021.*
>
> *[7] Noorani, Erfaun, et al. "From abstraction to reality: DARPA's vision for robust sim‐to‐real autonomy." AI Magazine 46.2 (2025): e70015.*
>
> > W3: Missing baselines
>
> Thanks for the suggestions. We would be more than willing to incorporate more baselines if the reviewer could kindly direct us to specific SOTA for analysis. We conducted a more comprehensive survey and provided detailed discussions in the Question section (Q2) for topics like context-based meta-learning, fine-tuning, and dynamic rescaling of actions. Besides, system identification is a classic but inefficient approach that requires estimating explicit dynamics parameters or even learning a full transition model, which typically demands large amounts of real-world data, whose procedures are slow, sensitive to noise, and often impractical for on-robot deployment.
>
>
> **Thank you for the precious opinions. We hope the explanation could resolve the reviewer's main concern, and looking forward to further discussions if needed, we will also incorporate all of the related work into the revised version of the paper at the end of the rebuttal.**

---

> > ### Comment · Reviewer_NFbW · 2025-11-27
> >
> > Thanks for the effort in addressing my comments. I have no further questions at this point. I am satisfied with the revision and the newly added results. I will raise my score.

---

> > > ### Author Response · Authors · 2025-12-01
> > > **Thanks for the recognition and raised valuation**
> > >
> > > Dear reviewer NFbW,
> > >
> > > We appreciate your timely feedback in rebuttal period and effort in reviewing our work. We are glad that our updated paper together with additional experiment, clarifications and analysis could fully address your concern.
> > >
> > > Thank you for the **`raised evaluation score of 6`**!
> > >
> > > Best,
> > > Authors.

---

### Official Review · Reviewer_141r · 2025-10-31

**Soundness:** 3
**Presentation:** 2
**Contribution:** 2
**Rating:** 4
**Confidence:** 4

**Summary:**

This paper proposes Found-Adapt, a latent adaptation framework for sim-to-real transfer of pre-trained foundation policies, extending the Hilbert-space foundation model (HILP; Park et al., 2024). The approach performs latent-space adaptation rather than full policy retraining. It first applies a cross-domain latent alignment via weighted regression to initialize a latent vector, updating the pre-trained model using a batch of target-domain data. Because this linear update approximates only a local warping of the latent space, the authors introduce a permutation-invariant MetaDynamic summary of target-domain states to encode dynamics differences, followed by a nonlinear adapter optimized for latent reward consistency. Experiments on locomotion tasks under simulated dynamics perturbations show improved transfer relative to direct deployment and domain randomization.

**Strengths:**

- The paper tackles the important problem of efficient sim-to-real adaptation for foundation policies, a critical and emerging area in RL.
- The method adapts policies through latent-space updates and avoids costly retraining.
- The paper is generally well organized and builds upon the prior HILP framework in a coherent way.
- The two modules (cross-domain alignment and nonlinear adapter) play distinct roles—one theoretically motivated, the other more practical and performance-driven.
- The paper compares against relevant baselines including direct deployment, domain randomization, and PAD.
- Results show consistent improvements over naive transfer and domain randomization, reaching similar performance to PAD.

**Weaknesses:**

While the paper is technically sound and addresses a timely topic, several issues limit its clarity and impact:

- Comparison to PAD:
  - The paper reports comparable performance to PAD but frames PAD as incomparably inefficient. However, PAD’s longer pre-training time is not directly comparable to Found-Adapt’s adaptation cost, since that cost corresponds to task pre-training rather than the adaptation method itself. PAD achieves similar reward recovery with only a few hundred online samples, compared to the 5k target samples used for Found-Adapt. The comparison would be more balanced if adaptation sample efficiency (rather than pre-training) were highlighted.
  - Although the architectures differ, both methods effectively correct representation drift between domains by learning mappings that preserve task-relevant latent consistency. PAD adapts the feature extractor online via self-supervision, while Found-Adapt adds a learned adapter that refines the latent vector offline. The similarity in effective mechanism could explain their comparable empirical performance.

- The paper does not analyze how the Hilbert embeddings change under dynamics perturbations and after adaptation. Since part of HILP’s original appeal lies in the interpretable geometry of the latent space, understanding how perturbations distort these embeddings—and whether adaptation restores that structure—would be a great addition. Without this analysis, it is unclear whether the adaptation preserves the interpretable properties of the original foundation policy framework.

- Unclear role of key components:
  - The linear realignment step largely reuses the HILP latent regression with added target data, but details such as the weighting parameter λ are not explained or analyzed.
  - The nonlinear MetaDynamic adapter is presented as a black box; its architecture, representation, and training details are missing, making it difficult to assess generality or reproducibility.
  - It is unclear whether, after a nonlinear adapter is added, the target alignment is still needed at all, or one could use the original (i.e. $\lambda = 0$) with similar performance.

- The number of gradient steps and sample efficiency of the adaptation process are not well described. Questions about overfitting or stability with different training and architecture remain open.
- The work builds heavily on Park et al. (2024), reusing the Hilbert-space policy and regression-based task prompting. The main novelty lies in the addition of the adapter, which, though reasonable, feels more heuristic than conceptually grounded.
- All tests are sim-to-sim under controlled perturbations; no real-robot experiments are shown, despite strong emphasis on sim-to-real transfer. This limits the external validity of the claims.

- Presentation issues:
  - Figure 5B is unreferenced.
  - Line 439 refers to a missing table ("Table ???").
  - The files in anonymous repository link are currently inaccessible, with message: “The requested file is not found.”

**Questions:**

1. What is the exact architecture, representation, dimensionality and training procedure of the MetaDynamic model? Is there prior work motivating this structure for cross-domain transfer?

2. How is the weighting parameter λ defined and tuned? Does its optimal value depend on target batch size?

3. What is the performance without the initial alignment ($\lambda = 0$, i.e. step 1 using only sim data)?

4. Could the proposed framework support online adaptation similar to PAD?

5. How do perturbations affect the Hilbert-space embeddings? Is there any analysis of representation drift between simulator and target domains?

---

> ### Author Response · Authors · 2025-11-25
> **Response to Reviewer 141r -  (1/5)**
>
> **We truly thank the reviewer 141r for the time and effort to review this work, and we are sorry for the oversight on details that might have caused the clarity concerns**.
>
> We'd like to respond to the Questions (Q) and Weaknesses (W) respectively with necessary discussions to resolve the potential confusion.
>
> ## Questions
>
> > Q1: What is the exact architecture, representation, dimensionality and training procedure of the MetaDynamic model? Is there prior work motivating this structure for cross-domain transfer?
>
> Thanks for pointing this out, we'd like to provide more details of the MetaDynamic module below:
>
> 1. **Architecture and Dimensionality**:
>
> MetaDynamic is a permutation-invariant set encoder inspired by the DeepSets [1]:
> It is constructed by three levels:
> `First`, an elementwise encoder $\phi_{\text{enc}}$ is an MLP (d-> 128->128, ReLU), `then` connected to a permutation-invariant pooling operator (mean pooling), `after that`, it connects to a projector $\rho$: MLP (128->128->64, ReLU) that produces dynamics representation invariant to both ordering and varying set size, and we have the overall computation of invariant representation $\eta$ as:
>
> $$
> \eta = \rho\left(
>     \frac{1}{N}
>     \sum_{j=1}^{N}
>         \phi_{\text{enc}}(\phi(\tilde{s}_j))
> \right)
> $$
>
> [1] Zaheer, Manzil, et al. "Deep sets." Advances in neural information processing systems 30 (2017).
>
> 2. **Representation**:
>
> The input is a set of target-domain state embeddings \{$\phi(\tilde{s}_j)$\} (j=1 ~ N) as in Eq.(4), and it comes from the frozen foundation-policy encoder. Then the MetaDynamic aggregates these representations into a distribution-level dynamics signature $\eta$ that captures how state-vistation statistics shift under different dynamics (latent dynamics description). E.g., As of a robot: under the same control policy, when the dynamics (ground friction, link mass, or actuator strength) change, the robot will likely visit different regions of the state space.
>
> 3. **Training procedure**:
>
> MetaDynamic is pre-trained offline using simulator trajectories sampled under 12 distinct dynamics configurations (4 gravity, 6 friction, and 2 actuator-strength perturbations).
> We use a hybrid objective:
> $$
> \mathcal{L} = \mathcal{L}_1 + \alpha \mathcal{L}_2,
> $$
>
> where $\mathcal{L}_1$ is the cross-entropy term that does not aim to enumerate all possible dynamics regimes, but instead provides a set of supervised anchor points that shape the latent space by teaching the encoder how dynamics-induced variations manifest in the observed state distributions. This supervision helps the model become sensitive to dynamic changes, while still allowing generalization to unseen dynamics during deployment. And $\mathcal{L}_2$ is the NT-Xent loss~[2] that pulls together sets sampled from the same dynamics and pushes apart sets sampled from different ones, yielding a smooth and discriminative dynamics signature. After pre-training, the encoder is frozen. During deployment, it only performs a forward pass to produce the dynamics signature from real target-domain state samples.
>
>
> > Q2: How is the weighting parameter $\lambda$ defined and tuned? Does its optimal value depend on target batch size?
>
> The $\lambda$ is a scalar that balances the influence of simulator data and target-domain data in the closed-form weighted regression. We tune $\lambda$ by grid search on a held-out split, using the range {0.5, 1.0, 2.0, 3.0}, and set $\lambda$ = 2.0 for tasks and all dynamics-shift regimes. The optimal value of $\lambda$ is independent of the target batch size. Because Eq. (3) operates on unnormalized squared residuals, so increasing the number of target samples only reduces the variance of the estimate but does not change the relative scale between the two terms. However, In our experiment, we acknowledge that the optimal value of $\lambda$ might change due to the different severity of the target domain shift and the sample quality of the target domain data, thus in application, parameter-tuning is still necessary for the optimal performance of transferability.
>
> > Q3: What is the performance without the initial alignment ($\lambda$ = 0, i.e. step 1 using only sim data)?
>
> Thanks for the question. $\lambda$ = 0 essentially reveals the performance of the model when no target-domain alignment is used, and the latent is derived purely from simulator data. We have the experiment across 4 tasks as in the “init” in our ablation study (Fig. 4a) corresponds exactly to the $\lambda$ = 0 case, where the latent is computed using only simulator data without any
> target-domain alignment. As shown in the figure, the performance drops across all tasks (especially in run and flip) it shows the importance of initial alignment step.

---

> ### Author Response · Authors · 2025-11-25
> **Response to Reviewer 141r -  (2/5)**
>
> > Q4: Could the proposed framework support online adaptation similar to PAD?
>
> Thanks for asking, and yes, our framework can support online adaptation, and in fact, it is naturally suited for this setting. Based on PAD [1], which requires performing a full gradient update of large CNN and policy networks at every environment step, this makes it `online` but `not real-time`, our method only updates a lightweight latent adapter via a closed-form solution and a MetaDynamic forward computation. This makes the adaptation cost negligible, which enables online updates with better real-time sense.
>
> [1] Hansen, Nicklas, et al. "Self-Supervised Policy Adaptation during Deployment." International Conference on Learning Representations.
>
> > Q5: How do perturbations affect the Hilbert-space embeddings? Is there any analysis of representation drift between simulator and target domains?
>
> Thanks for the question. We want to clarify that dynamics perturbations do not modify the Hilbert-space encoder itself. The encoder $\phi(\cdot)$ is pretrained as part of the foundation model and is frozen during the adaptation stage. Therefore, the mapping $s \mapsto \phi(s) \in \mathbb{R}^D$ would be identical given the same batch of input.
>
> However,  **we agree with reviewer that the embeddings $\phi(s)$ will differ when the input states $s$ come from different target domains**. Dynamics
> perturbations alter the underlying state-visitation distribution: $p_{\text{sim}}(s) \neq p_{\text{targ}}(s)$, and thus the embeddings $\phi(s)$ inevitably shift. It can be recognized that, this shift reflects how different dynamics lead the pretrained HILP encoder to understand the state space differently, which then propagates into the degradation of the inferred latent $z$ that eventually leads to the policy performance drop.
>
> To verify this, we collected the samples from five varied target domain settings in Walk task, (one default source domain, and four target domains), and present the 2D t-SNE figure showing the $\phi(s)$, as in **Appendix, Fig. 7**. Under the default gravity $g_0$, the Hilbert-space embedding forms a clean, well-organized manifold. As gravity increases ($g_1 \rightarrow g_4$), the encoder produces progressively more distorted and contracted representations, because with the same action, the agent possibly explores different parts of the state space and experiences different transition dynamics, which illustrates the existence of the shift. Due to this, the policy that was constructed based on the Hilbert-space expressions would not be able to tackle the tasks in the target domain successfully, and that's why it motivates us to perform the policy adaptation.

---

> ### Author Response · Authors · 2025-11-25
> **Response to Reviewer 141r -  (3/5)**
>
> ## Weaknesses
>
> > W1: Comparison to PAD:
>
> Thanks for raising this point. We'd like to provide more discussion. The goal of Found-Adapt is fundamentally different from PAD, which naturally leads to different training and sample-use characteristics.
>
> **1. Task-agnostic vs. task-specific adaptation.**
> PAD performs *task-specific* self-supervision: for each downstream task, PAD needs to update the encoder and policy parameters. In contrast, Found-Adapt builds upon a **task-agnostic foundation representation** learned by HILP. A *frozen encoder* is shared across all tasks, and our method adapts only a lightweight latent vector. This allows adaptation to be shared across multiple behaviors (e.g., walk, flip, etc) without re-training a network per task. The flexibility of task-agnostic adaptation is a key design objective of Found-Adapt and differs from PAD’s per-task supervision.
>
> **2. Different roles of target-domain samples.**
>
> The target-domain transitions in our setting are used to calibrate a shared latent representation that applies across multiple behaviors. It is to provide a task-agnostic estimate of the domain shift rather than to train a task-specific policy.
>
> In contrast, PAD uses target-domain samples to update the feature extractor and policy for each behavior task. Because the learning objectives differ (task-agnostic latent alignment vs. per-task encoder adaptation), the absolute number of target samples used by the two methods is not directly comparable. Our design focuses on leveraging target data to estimate domain-level changes once, enabling reuse across walk, stand, run, and flip without learning a separate adaptation model for each task.
>
> To better highlight this difference, we have added clarifications in the revision Section 5.1 and Appendix , including a discussion of task-agnostic reuse and the role of target-domain samples.
>
>
> > W2:  Analyze how the Hilbert embeddings change under dynamics perturbations and after adaptation.
>
> Thanks for the thoughtful comment. We clarify that there is a `misunderstanding` about the role of Hilbert embeddings in our framework. In our work, the Hilbert-space encoder $\phi(\cdot)$ from HILP is kept frozen during the adaptation. Therefore, the embedding outcome $\phi(s)$ differs across dynamics settings, this difference comes from the state-distribution shift (as shown in Fig. 7).
>
> Since our method does not alter $\phi$, comparing Hilbert embeddings “before vs. after adaptation’’ would not be meaningful in our setting: the embeddings remain unchanged given the same batch of input, and adaptation cannot (and is not intended to) reshape the Hilbert
> manifold. Instead, what changes is the *latent representation* $z$ used by the policy. We have such study analysis and want to kindly direct the reviewer to **The visualization in Fig. 13**: the top row shows the initial latent $z_{\text{src}}$, which performs poorly under the target-domain dynamics, while the bottom row shows $z_{\text{refined}}$, where the adapter gradually adjusts the latent over the adaptation steps. As the loss decreases, the latent converges to a more appropriate representation for the perturbed domain, leading to improved returns in $E_{\text{real}}$.
>
> > W3: Unclear role of key components.
>
> Since there is an overlap, we merge the answers to the questions section as follows:
>
> 1. Weighting parameter $\lambda$: Q2
> 2. MetaDynamic details: Q1
> 3. Performance without the initial alignment ($\lambda=0$): Q3
>
> > W4: The number of gradient steps and sample efficiency of the adaptation process are not well described. Questions about overfitting or stability with different training and architecture remain open.
>
> Thanks for the question. In our method, only a small adapter MLP $g_\theta$ is updated during adaptation; the Hilbert encoder and MetaDynamic network all remain. The optimization is therefore very lightweight: we apply a fixed number of Adam steps (300 in our setting) on a low-dimensional latent vector ($D=32$) under the simple regression loss $\mathrm{MSE}(\Phi_{\text{tar}} g_\theta(z_{\text{src}},\eta), r_{\text{targ}})$ as in Eq (5). Because no large network or complex network architecture is finetuned, the risk of overfitting or instability is minimal. Besides, Figure 13 shows that the loss decreases smoothly and the latent converges stably. It confirms that the adaptation is reliable in practice. We have added a clarification of the details in the revision Caption of Figures 13, Line 1037-1039.

---

> ### Author Response · Authors · 2025-11-25
> **Response to Reviewer 141r - (4/5)**
>
> > W6: Sim-to-sim setting:
>
> Thanks for the concern. Our work focuses on foundation-policy based dynamics adaptation, which requires **controlled and repeatable variations in system dynamics** in order to isolate and evaluate the effectiveness of latent-space refinement. In addition, we follow the standard sim-to-sim evaluation protocol, which is widely adopted in prior work [1][2][3] to serve as the established way to assess sim-to-real adaptation performance with better reproducibility. We appreciate the reviewer's understanding and will include the physical experiment in the follow-up work.
>
> [1] Eysenbach, B., Chaudhari, S., Asawa, S., Levine, S., & Salakhutdinov, R. Off-dynamics reinforcement learning: Training for transfer with domain classifiers. In International Conference on Learning Representations (ICLR).
>
> [2] Hansen, Nicklas, and Xiaolong Wang. "Generalization in reinforcement learning by soft data augmentation." 2021 IEEE International Conference on Robotics and Automation (ICRA). IEEE, 2021.
>
> [3] Noorani, Erfaun, et al. "From abstraction to reality: DARPA's vision for robust sim‐to‐real autonomy." AI Magazine 46.2 (2025): e70015.
>
> > W7: Presentation Issues:
>
> Thank you very much for the detailed suggestions. We are very sorry for the minor mistakes, and we have revised all three writing oversights. Besides, we have conducted a thorough proofread and made sure the uploaded version is complete and error-free.

---

> ### Author Response · Authors · 2025-12-01
> **Response to Reviewer 141r - (5/5)**
>
> ### Additional Experiment Results for `Q2` (λ = 0) and `Q3`  (λ = {0.5, 1, 2, 3}):
>
> To provide further details and support the discussions for Q2 and Q3 regarding the question of the parameter $\lambda$, we conducted experiments with 5 different values on two tasks (`stand`, `walk`) for demonstration, the results are shown as below, where λ = 0 means there is no target-domain alignment used, and λ = 2 is the selected value in this work, in this experiment, we could observe that, when there is no target domain information used for adaptation, the results are consistently worse than the current result, this is because the initial alignment fails to bridge the gap from the first step of closed-form solution, besides, the other parameter settings reveals the sensitiveness of the sim-to-real adaptation performance to the λ, in general, incorporating more target-domain evidence improves latent correction, but overly large values start to overweight noisy or limited target samples:
>
> Task: **Stand**
>
> |λ| g1| g2| g3| g4|
> |-------|---------|----------|----------|----------|
> | 0.0     | 548.96  | 220.34   | 168.43  | 30.24   |
> | 0.5   | 543.99  | 219.11   | 135.46 | 29.49   |
> | 1.0     | 536.98  | 235.54   | 148.01   | 81.87   |
> | 2.0     | **586.63**  | **276.62**   | **276.15**  | **76.70**   |
> | 3.0     | 553.02  | 206.58   | 233.87  | 29.78   |
>
> Task: **Walk**
>
> | λ   | g1      | g2      | g3      | g4      |
> |-----|---------|---------|---------|---------|
> | 0.0 | 202.45  | 157.15   | 95.37   | 41.03   |
> | 0.5 | 158.34  | 161.91   | 97.36   | 29.16   |
> | 1.0 | 369.67  | 163.50   | 92.08   | 32.61   |
> | 2.0 | **472.25** | **182.46** | **120.87** | **118.55** |
> | 3.0 | 439.78 | 180.96   | 98.00   | 28.62   |
>
>
>
> Once again, `we appreciate the constructive feedback from reviewer 141r`. Your detailed feedback makes the work more robust and comprehensive, from the experimental details, presentation skills, and detailed clarifications. `All of the revisions are made to the new version of PDF`.
>
> Due to the current policy, **we regret that further interactive discussion is not feasible, but we truly hope the clarifications, additional experiments, and detailed analysis could fully address your concerns**. We sincerely value your time and expertise, and we thank you again for helping us improve the work.

---

### Official Review · Reviewer_QoG5 · 2025-11-01

**Soundness:** 3
**Presentation:** 4
**Contribution:** 2
**Rating:** 4
**Confidence:** 4

**Summary:**

This paper addresses the critical challenge of sim-to-real transfer for reinforcement learning policies. The authors propose "Found-adapt," a framework that builds upon foundation policies (specifically, the HILP[1] framework) to enable efficient adaptation from a source simulator to a target domain with different dynamics. The core idea is to decouple skill acquisition (pre-trained in sim) from environment adaptation via a task vector $z$ in the Hilbert task space.

The proposed adaptation mechanism consists of three stages. First, a **Cross-Domain Initial Alignment** step computes an initial latent vector $z_{src}$ via a weighted least-squares regression from both sim and real data. Second, a **Dynamics Signature** is generated using a pre-trained, permutation-invariant network to encode a "dynamics signature" $\eta$ from a small batch of target-domain data. Finally, in the **Signature-Guided Adaptation** stage, a small adapter network, $g_{\theta}$, takes both $z_{src}$ and $\eta$ as input is then fine-tuned for a small number of steps on the target-domain to get the final, adapted latent vector $z_{final}$, which is subsequently used by the frozen foundation policy.

The authors evaluate their method on several "sim-to-sim" locomotion tasks (e.g., Walker, Cheetah) with varying dynamics (friction, gravity). The results show that Found-adapt can mitigate the performance gap and offers a parameter-efficient alternative to retraining the entire policy.

**Strengths:**

* Important Problem: The paper tackles the sim-to-real gap, which remains a significant and high-impact obstacle for the practical application of reinforcement learning in robotics and the real world.
* Promising Framework Idea: The core concept of adapting the *latent space* of a foundation policy, rather than fine-tuning the policy network itself, is appealing. It suggests a modular, parameter-efficient, and fast adaptation method.
* Conceptually Sound Decomposition: The idea of decomposing the adaptation problem into an initial task-aligned latent ($z_{src}$) and a separate dynamics signature ($\eta$) is interesting. It attempts to distinguish "what to do" from "how the environment behaves."

**Weaknesses:**

* Clarity of Contribution (vs. HILP): **(major)** The paper dedicates a very large portion of its methodology (Sec 4.1) to a detailed recapitulation of the HILP framework [1], which is existing prior work. This positioning obscures the paper's own, new contributions. The actual novel parts are confined to Sec 4.2. Of these, Sec 4.2.A (weighted regression) is a straightforward and minor extension of HILP's original least-squares solver. This leaves Sec 4.2.B/C as the main contribution, however, the `MetaDynamic` network (Sec 4.2.B), is not clearly explained. The paper states it is "trained on simulator data and frozen at deployment" but provides no details on its training objective, loss function, architecture, or the data used for its pre-training. This is a major omission that makes the contribution difficult to assess and the work impossible to reproduce.

* Weak Experimental Validation (Sim-to-Sim): The paper's abstract and introduction frame the problem as "real-world application" and "sim-to-real transfer." However, all experiments are conducted in a "sim-to-sim" setting (transferring between different MuJoCo physics parameters). While this is a common practice, the lack of a single real-world hardware experiment significantly weakens the paper's central claims.

* Marginal Empirical Results: The empirical results are not strongly convincing. In Table 1 and Figure 3, the proposed method ("Ours") shows only marginal improvements over the baselines. In some cases (e.g., Stand and Flip tasks under G3/G4 gravity), the "Ours" method appears to be outperformed by the PAD baseline, suggesting it may not be uniformly more robust.
* Missing Related Work (RMA): The "Adaptive Policy Networks" related work section is notably incomplete. It omits the highly relevant and successful line of work on Rapid Motor Adaptation (RMA) like[2-5], which also uses a latent-conditioned policy ($z$) to adapt to different dynamics.

* Missing Critical Ablation (Direct Latent Optimization): The paper's primary claim is that its ($z_{src}$, $\eta$) $\rightarrow$ $g_{\theta}$ $\rightarrow$ $z_{final}$ pipeline is an effective adaptation mechanism. However, it fails to compare against a much simpler and highly relevant baseline: directly optimizing the latent vector $z$ itself using data from the target environment. This approach (treating $z$ as a small set of learnable parameters at deployment) has shown great success in related work (e.g., RTR [5]), which also adapts a latent-conditioned policy for sim-to-real). It is unclear if the proposed `MetaDynamic` network and adapter $g_{\theta}$ are necessary at all, or if simply fine-tuning $z_{src}$ would achieve the same or better results.

**Questions:**

The main questions are elaborated in the weakness part, here are some minor questions:

* In the ablation study (Fig 4a), the `F(init+dyna)` variant performs very poorly. This implies the adapter $g_{\theta}$ is not just combining $z_{src}$ and $\eta$ but reconciling them. Could the authors explain why this simple combination fails so badly and what this reveals about the learned representations of $z_{src}$ and $\eta$?
* How was the hyperparameter $\lambda$ (the weight for target-domain data in Sec 4.2.A) selected, and how sensitive is the performance of $z_{src}$ (and $z_{final}$) to this choice?
* How sensitive is the framework to the amount of target-domain data ($M$)? What is the minimum number of target samples required for the `MetaDynamic` network to extract a useful signature $\eta$ and for the adapter $g_{\theta}$ to be effectively tuned?

---

> ### Author Response · Authors · 2025-11-22
> **Response to Reviewer QoG5**
>
> **We appreciate the time and effort of reviewer QoG5 for providing us constructive feedbacks.**
>
> We'd like to provide detailed explanations to each of the Weakness and Question.
>
> > W1: Clarity of Contribution
>
> `1. Our method innovates from HILP foundation policy backbone in following aspects`:
>
> - In Sec.4.2A, we correct latent inference under dynamics shift by integrating simulator and real-domain features in a weighted closed-form solver. However, because $\Phi_{tar}$ only captures local transition differences and becomes increasingly limited in high-dimensional state spaces, we admit relying solely on this closed-form step cannot account for the broader structural or distributional changes induced by dynamics shift.
> - Thus, In Sec.4.2B, we introduce a permutation-invariant network that extracts higher-order dynamics signatures from target-domain states.
> - In Sec.4.2C, we refine the latent vector using a parameter-efficient adapter conditioned on the dynamics signature $\eta$.
>
> **In contrast**, HILP performs no latent refinement, no dynamics modeling, and no adaptation at deployment, it assumes the $\Phi_{sim} = \Phi_{real}$, which essentially breaks under sim-to-real conditions and our method intends to solve this generalizable problem only by leveraging the foundation model's backbone.
>
> We included Sec. 4.1 to establish the necessary background and notation for the foundation-policy setting, ensuring that all readers share a common understanding before introducing our method. We will condense this section to highlight the contributions of the work.
>
>
> `2. Detailed explanations of MetaDynamic network (Sec 4.2.B)`:
>
> **The MetaDynamic is designed to encode a set of target-domain dynamics reflected from the collected samples**. These samples might derived from different rollouts, and it is the distributional pattern that actually matters (e.g., contact-phase frequency, or velocity spread, etc, influenced by the system dynamics), so, the encoder should be permutation-invariant, and produce the consistent representation regardless of how the samples are ordered, meanwhile, most importantly: capturing the underlying statistics that reflect the dynamics characteristics.
>
> In our implementation, the MetaDynamic network contains an element-wise MLP $\phi(\cdot)$ that embeds each state independently, then a mean-pooling operator that aggregates the set into a fixed-dimensional summary vector, and another second MLP $\rho(\cdot)$  that produces the dynamics signature with set-level invariant. We will incorporate these clarifications and architectural details into the revised version. This design is inspired by [1], which provides the theoretical guarantee to stably capture the target-domain dynamics distribution.
>
>
>
> `3. More details including training objective, loss function, architecture, and the data used for its pre-training of MetaDynamic network`:
>
> The MetaDynamic module is a permutation-invariant set encoder that maps a small set of target-domain encoded states into a compact dynamics signature. As shown in Equation below,  it follows the DeepSets formulation by applying an elementwise encoder, a permutation-invariant pooling operator, and a global projector. The architecture and training procedure (including simulator dynamics variations) are now fully documented in Appendix G.
>
> $$
> \eta = \rho\\left(
>     \frac{1}{N} \sum_{j=1}^{N} \phi_{\text{enc}}(\phi(\tilde{s}_j))
> \right)
> $$
>
> During pre-training, the encoder is trained offline using simulator trajectories under 12  different dynamics configurations (gravity, friction, and actuator-strength perturbations).  We use a hybrid objective combining a dynamics-classification term and a contrastive term:
>
> $\mathcal{L} = \mathcal{L}_1 + \alpha\mathcal{L}_2$, where  $\mathcal{L}_1$ is the cross-entropy term that encourages the encoder to distinguish simulated dynamics regimes, and  $\mathcal{L}_2$ is the NT-Xent loss [2] pulls together sets sampled from the same  dynamics and pushes apart sets sampled from different ones, yielding a smooth and  discriminative dynamics signature. After pre-training, the encoder is frozen. During  deployment, it only performs a forward pass to produce the dynamics signature from real  target-domain state samples.
>
>
> [1] Zaheer, Manzil, et al. "Deep sets." Advances in neural information processing systems 30 (2017).
>
> [2] Chen, Ting, et al. "A simple framework for contrastive learning of visual representations." International conference on machine learning. PmLR, 2020.

---

> ### Author Response · Authors · 2025-11-22
> **Response to Reviewer QoG5 - Continue**
>
> > W2: Sim-to-Sim Concern
>
> Thanks for the concern, Our work focuses on foundation-policy–based dynamics adaptation, it requires controlled and repeatable variations in system dynamics to evaluate the effectiveness of latent-space refinement. Besides, we follow the sim-to-sim, which is a widely adopted protocol as in work [3][4][5], to verify the sim-to-real performance.
>
> [3] Eysenbach, B., Chaudhari, S., Asawa, S., Levine, S., & Salakhutdinov, R. Off-dynamics reinforcement learning: Training for transfer with domain classifiers. In International Conference on Learning Representations (ICLR).
>
> [4] Hansen, Nicklas, and Xiaolong Wang. "Generalization in reinforcement learning by soft data augmentation." 2021 IEEE International Conference on Robotics and Automation (ICRA). IEEE, 2021.
>
> [5] Noorani, Erfaun, et al. "From abstraction to reality: DARPA's vision for robust sim‐to‐real autonomy." AI Magazine 46.2 (2025): e70015.
>
>
> > W3: The empirical results are not strongly convincing. In some cases (e.g., Stand and Flip tasks under G3/G4 gravity), the "Ours" method appears to be outperformed by the PAD baseline, suggesting it may not be uniformly more robust.
>
> In Table 1 and Figure 3, the proposed method ("Ours") shows consistent performance of best or second best (underline), besides, the few cases where PAD appears higher are those where PAD conducts full on-target training for each dynamics environment, whereas our method performs zero-training adaptation via a latent adjustment. As in Table 1, the advantage of the proposed Found-adapt is that it consumes minimal cost time while maintaining good sim-to-real performance.
>
> > W4: Missing Related Work (RMA): The "Adaptive Policy Networks" related work section is notably incomplete. It omits the highly relevant and successful line of work on Rapid Motor Adaptation (RMA) like[2-5], which also uses a latent-conditioned policy to adapt to different dynamics.
>
> We are very grateful for reviewer's suggestion and remind for more comprehensive literature review. However, the references denoted as “`[2–5]`” did not appear in the review from reviewer. `We wonder could the reviewer kindly provide the specific citations intended from [1] to [5]`, we would be very happy to incorporate and discuss them in detail.

---

> > ### Comment · Reviewer_QoG5 · 2025-11-23
> >
> > Thank the authors for the clarification. I would appreciate it if the method section include more details about the MetaDynamic and maintain more concisely on the HILP part. I'm sorry about the oversight in not including the references:
> > I'll raise my score to 6 in the hope that these concerns will be addressed in the final version.
> > **Reference**
> >
> > [1] Park, Seohong et al. “Foundation Policies with Hilbert Representations.” *ArXiv* abs/2402.15567 (2024): n. pag.
> >
> > [2] Kumar, Ashish, et al. "Adapting rapid motor adaptation for bipedal robots." *2022 IEEE/RSJ International Conference on Intelligent Robots and Systems (IROS)*. IEEE, 2022.
> >
> > [3] Qi, Haozhi, et al. "In-hand object rotation via rapid motor adaptation." *Conference on Robot Learning*. PMLR, 2023.
> >
> > [4] Liang, Yichao, Kevin Ellis, and Joao Henriques. "Rapid motor adaptation for robotic manipulator arms." *Proceedings of the IEEE/CVF Conference on Computer Vision and Pattern Recognition*. 2024.
> >
> > [5] Hu, Kaizhe, et al. "Robot Trains Robot: Automatic Real-World Policy Adaptation and Learning for Humanoids." *arXiv preprint arXiv:2508.12252* (2025).

---

> ### Author Response · Authors · 2025-11-23
> **Thank you for the raised evaluation**
>
> > Q1: `F(init+dyna)` performance analysis and understanding
>
> Thanks for the question. The performance of the `init+dyna` variant indeed shows that the adapter is not merely combining $z_{\text{src}}$ and $\eta$, but reconciling them. These two vectors encode different types of information and exist in different coordinate systems: I.e., $z_{\text{src}}$ captures Hilbert embeddings, while $\eta$ summarizes domain-level dynamics statistics. A naive concatenation disrupts the geometry learned during foundation pretraining, producing a latent vector that no longer respects the structure required for meaningful policy behavior. In contrast, the adapter learns a structured latent-space transformation that aligns the dynamics signature with the Hilbert embedding, thus corrects dynamics-induced drift.
>
> > Q2: How was the hyperparameter $\lambda$ selected?
>
> Thanks for the question, We tune $\lambda$ by grid search on a held-out split, using the range {0.5, 1.0, 2.0, 3.0}, and set
> $\lambda$ = 2.0 for tasks and all dynamics-shift regimes. We also provided details sensitive analysis in the **Response to Reviewer 141r - (4/4)**. We kindly direct the reviewer to check more discussion in the `Q2 of Reviewer 141r` and the added experiment section.
>
> > Q3: How sensitive is the framework to the amount of target-domain data?
>
> Thanks for the question. Our sensitivity analysis in Fig. 5(a) directly evaluates how performance changes when the target-domain data are progressively reduced or degraded through three corruption modes (drop, mask, noise). All three modes remain stable when corruption is less than 50%-75%, it shows that the method can adapt reliably even when only a modest portion of the target transitions is available. We have a more comprehensive discussion in line 438 under *(4) The analysis of the requirement of target-domain data*.
>
>
>
>
> **We appreciate your constructive feedback and the detailed reference list for us to better position our work on the existing effort**. Your suggestions are well received and we will keep working on the updated version with highlighted changes to reflect the revisions discussed in the rebuttal!
>
> **Thank you again for the positive feedback and `raised evaluation score of 6`**!

---

### Meta-Review · Area_Chair_5VBs · 2026-01-07

**Summary:**

This paper proposed a new parameter-efficient sim-to-real adaptation method for foundation policies, by operating in a learned latent/Hilbert representation space via cross-domain least-squares latent alignment. The paper studies an important problem that is very well-motivated, and the experimental results are solid and thorough, with strong practical appeal and, in particular, fast adaptation relative to stronger baselines. However, there were also concerns regarding the realism of the evaluation, the clarity/justification of the method, as well as the stability and identifiability issues of the learned dynamics. The rebuttal has adequately addressed most of the concerns, and two of the reviewers have explicitly raised the score, making the paper clear the bar for ICLR. I recommend that the authors incorporate the feedback in preparing the camera-ready version of the paper.

**Reviewer Concerns:**

Concerns regarding the missing of some related work and better positioning of the paper, the insufficiency of some experiment details, and clarifications on the technical novelty, have been properly addressed. There are still some concerns regarding real-robot experiments, the sufficiency of the comparison with particular baselines, and the clarity of the proposed method, but not significantly outstanding.

**Reviewer Scores:**

Reviewers QoG5 and NFbW have already explicitly increased their scores from 4 to 6 after the rebuttal, after discussions with the authors. Reviewer A9Lz will likely to stay at 8, and Reviewer 141r will likely to increase the score as well, as their major concern was on the missing details of the implementation, which has been addressed.

---

### Decision · Program_Chairs · 2026-01-26

Accept (Poster)